# B cell receptor and Toll-like receptor signaling coordinate to control distinct B-1 responses to both self and the microbiota

Lieselotte SM Kreuk[1], Meghan A Koch[1†], Leianna C Slayden[1], Nicholas A Lind[1], Sophia Chu[1], Hannah P Savage[2], Aaron B Kantor[3], Nicole Baumgarth[2], Gregory M Barton[1]*

[1]Department of Molecular and Cell Biology, University of California, Berkeley, Berkeley, United States; [2]Center for Comparative Medicine, University of California, Davis, Davis, United States; [3]Department of Genetics, Stanford University, Stanford, United States

*For correspondence:
barton@berkeley.edu

Present address: †Division of Basic Sciences, Fred Hutchinson Cancer Research Center, Seattle, United States

Competing interests: The authors declare that no competing interests exist.

**Abstract** B-1a cells play an important role in mediating tissue homeostasis and protecting against infections. They are the main producers of 'natural' IgM, spontaneously secreted serum antibodies predominately reactive to self antigens, like phosphatidylcholine (PtC), or antigens expressed by the intestinal microbiota. The mechanisms that regulate the B-1a immunoglobulin (Ig) repertoire and their antibody secretion remain poorly understood. Here, we use a novel reporter mouse to demonstrate that production of self- and microbiota-reactive antibodies is linked to BCR signaling in B-1a cells. Moreover, we show that Toll-like receptors (TLRs) are critical for shaping the Ig repertoire of B-1a cells as well as regulating their antibody production. Strikingly, we find that both the colonization of a microbiota as well as microbial-sensing TLRs are required for anti-microbiota B-1a responses, whereas nucleic-acid sensing TLRs are required for anti-PtC responses, demonstrating that linked activation of BCR and TLRs controls steady state B-1a responses to both self and microbiota-derived antigens.
DOI: https://doi.org/10.7554/eLife.47015.001

## Introduction

B-1a cells were discovered 35 years ago and have characteristics that bridge the innate and adaptive immune system (*Herzenberg et al., 1986*). Unlike follicular B cells, termed B-2, B-1a cells are rapidly recruited to sites of infection and produce antibodies independently of T cell help. B-1a cells are the main producers of serum IgM antibodies (*Lalor et al., 1989a*; *Baumgarth et al., 1999*; *Ohdan et al., 2000*; *Haas et al., 2005*; *Choi and Baumgarth, 2008*; *Holodick et al., 2009*), which promote tissue homeostasis and provide protection against infections (*Haas et al., 2005*; *Choi and Baumgarth, 2008*; *Boes et al., 1998*; *Ochsenbein et al., 1999*; *Boes et al., 2000*; *Ehrenstein et al., 2000*; *Baumgarth et al., 2000*; *Alugupalli et al., 2003*; *Notley et al., 2011*; *Vas et al., 2013*). More recently B-1a cells have been implicated as a source of microbiota-reactive, class-switched IgG and IgA antibodies, which are important for intestinal homeostasis (*Kroese et al., 1989*; *Kroese et al., 1993*; *Lalor, 1991*; *Kroese et al., 1996*; *Bos et al., 1996*; *Macpherson et al., 2000*; *Koch et al., 2016*; *Savage et al., 2017*).

B-1a cell-derived antibodies are often reactive with self epitopes (*Hayakawa et al., 1999*; *Mercolino et al., 1988*; *Yang et al., 2015*). For example, the most common B-1a specificity is for phosphatidylcholine (PtC), a phospholipid present within the plasma membranes of eukaryotic cells

(*Mercolino et al., 1988*). Other specificities include nucleic acids, LPS, and carbohydrates. Early studies from Hayakawa and colleagues showed that the absence of a specific self-antigen, such as Thy-1, results in the absence of self-reactive B-1a specificities to that antigen (*Khan et al., 1995a*). Moreover, mice harboring mutations in BCR signaling molecules such as Btk exhibit a reduction in B-1a B cell numbers (*Khan et al., 1995b*), whereas mice with deficiencies in BCR coinhibitory molecules such as CD72 have increased frequencies of B-1a cells (*Pan et al., 1999*). These studies strongly suggest that positive selection via BCR signaling is critical for B-1a cell development and/or maintenance.

Many B-1a derived antibodies have been termed polyreactive due to their ability to bind shared structures present on a variety of apparently unrelated self and foreign antigens. For example, B-1a-derived monoclonal antibodies can recognize epitopes present on cell membranes of apoptotic cells but also on bacterial cell wall polysaccharides (*Boes et al., 2000*; *Ehrenstein et al., 2000*; *Notley et al., 2011*; *Nguyen et al., 2015*). A recent study by *Yang et al. (2015)* reported no significant difference in the immunoglobulin heavy chain repertoire of B-1a cells from adult germ free (GF) and specific-pathogen-free (SPF) mice, supporting a model whereby the B-1a repertoire is not defined by microbiota-derived antigens. Consequently, B-1a antibody responses directed against the microbiota are generally thought to arise from BCR specificities selected on self but cross-reactive to microbial antigens. However, the extent to which self versus microbiota-derived antigens contribute to the clonal expansion or antibody secretion of B-1a cells remains unknown.

B-1a derived serum antibodies are termed 'natural' IgM due to their constitutive secretion, which is thought to be independent of antigen exposure (*Ochsenbein et al., 1999*; *Baumgarth et al., 2015*). Additionally, whereas the B-2 cellular compartment is maintained by BM-derived hematopoietic cells (HSCs), B-1a cells are primarily of fetal and neonatal origin and are maintained by self-renewal throughout life (*Lalor et al., 1989a*; *Lalor et al., 1989b*; *Montecino-Rodriguez et al., 2006*; *Yoshimoto et al., 2011*; *Kobayashi et al., 2014*; *Hardy and Hayakawa, 2015*). Unlike B-2 cells, in vitro ligation of the BCR with anti-IgM on mature B-1a cells leads to reduced calcium mobilization and induction of apoptosis (*Morris and Rothstein, 1993*; *Bikah et al., 1996*; *Ochi and Watanabe, 2000*). The inability of mature B-1a cells to proliferate in response to BCR-mediated stimulation in vitro has been attributed to their expression of CD5, a negative regulator of TCR and BCR signaling (*Bikah et al., 1996*; *Azzam et al., 1998*; *Hippen et al., 2000*; *Perez-Villar et al., 1999*). These observations have led to a model whereby B-1a cell development is mediated by positive selection on self antigens, whereas activation and antibody production is regulated by nonclonal activation through pattern recognition receptors, such as Toll-like receptors (TLRs) or cytokine receptors (*Nisitani et al., 1995*; *Waffarn et al., 2015*; *Ha et al., 2006*). Indeed, B-1a cells express a variety of TLRs (TLR1, 2, 3, 4, 7, 8, and 9) and are more prone to terminal plasma cell differentiation than B-2 cells upon in vitro stimulation by TLR ligands (*Genestier et al., 2007*; *Meyer-Bahlburg et al., 2009*). B-1a cell-intrinsic TLR signaling has also been implicated in the egress of B-1a cells from the peritoneal cavity to the spleen and intestinal sites following challenge (*Ha et al., 2006*; *Murakami et al., 1994*). However, studies investigating TLR-dependent B-1a responses have largely been performed either in the context of infection or in vitro. Thus, the contribution of TLR signaling to steady state in vivo B-1a responses remains unclear.

In addition to positive selection, there is evidence that B-1a cells receive intermittent or ongoing BCR signaling throughout life, leading to their activation and expansion. A high frequency (15–30% of peritoneal cavity and 10–15% of splenic) of PtC-reactive B-1a cells is consistently observed in all strains of inbred mice, a significant percentage of which express the canonical VH11-2/VK14 heavy and light chain gene pair. Additionally, while young mice have a more diverse B-1a repertoire, dramatic BCR restriction occurs as animals age (*Yang et al., 2015*). This restriction suggests an antigen-specific BCR-mediated expansion process. In support of this model, a recent study from Zikherman and colleagues found that B-1a cells express high levels of Nur77 (*Huizar et al., 2017*), a gene associated with BCR signaling. Therefore, it is likely that B-1a cells are indeed receiving antigen specific BCR signaling in vivo, which may be important for their activation and expansion. However, it is still unclear how B-1a cells are able to respond to BCR-mediated activation, given their inherent inability to response to anti-IgM stimulation in vitro and expression of negative regulators of BCR signaling such as CD5.

In the present study we investigate the contribution of BCR and TLR signaling in regulating B-1a activation and function. We use a novel reporter mouse to show that the generation of self- and

microbiota-reactive antibodies is linked to BCR signaling in B-1a cells. Moreover, we show that Toll-like receptors (TLRs) are critical not only for antibody secretion but also for defining the steady state B-1a immunoglobulin repertoire. Specifically, we find that nucleic acid-sensing TLRs are required for anti-PtC responses, whereas both a microbiota and bacterial-sensing TLRs are required for anti-commensal responses. Altogether, we propose a model whereby dual BCR and TLR signaling is required for the maintenance and antibody secretion of steady state B-1a responses to both self and gut microbial antigens, which may be important for preventing inappropriate responses leading to auto-immunity while aiding in the maintenance of tissue and intestinal homeostasis.

## Results

### Ighg3 reporter mouse marks a subset of activated, antibody secreting B-1a cells

Previous findings from our group showed that healthy mice make class-switched IgG3 and IgG2b serum antibodies reactive with the microbiota, which coordinate with IgA to dampen neonatal mucosal T cell responses (*Koch et al., 2016*). In order to better understand the origin and fate of IgG3$^+$ B cells we created a knock-in mouse in which IgG3$^+$ B cells express Cre recombinase (Cre) (*Figure 1A*). A targeting vector was used to introduce *T2A-Cre* followed by a *frt*-flanked *neomycin-resistance* gene into the *Ighg3* constant region, just after the exon encoding the last *Ighg3* trans-membrane domain (*Figure 1—figure supplement 1A*). This design should link expression of Cre to translation of IgG3 protein. Southern blotting confirmed correct targeting of the *Ighg3* locus (*Figure 1—figure supplement 1B*). We also confirmed a single insertion into the genome by southern blotting for the *neomycin-resistance* gene (*Figure 1—figure supplement 1C*). *Ighg3*$^{T2A-Cre}$ mice were crossed to *β-actin*$^{Flippase}$ mice to delete the *neomycin-resistance* gene. The resulting *Ighg3*$^{T2A-Cre}$ mice were intercrossed to *Rosa26*$^{STOPflox-TdTomato}$ reporter mice to generate mice homozygous for *Ighg3*$^{T2A-Cre}$ and *Rosa26*$^{STOPflox-TdTomato}$, hereafter referred to as *Ighg3*$^{T2A-Cre:TdTomato}$. In these mice the Tomato fluorescent protein should be expressed in any cell that expresses or has previously expressed IgG3 protein.

In adult *Ighg3*$^{T2A-Cre:TdTomato}$ mice, Tomato expression was restricted to CD19$^+$ B cells in the spleen (*Figure 1B*). However, we found that only a small fraction of Tomato$^+$ cells were IgG3$^+$ (*Figure 1B*). Unexpectedly, over half of Tomato$^+$ splenic B cells were IgG3$^-$IgM$^+$ (*Figure 1B*). We considered three possible mechanisms that could lead to the high frequency of IgG3$^-$IgM$^+$Tomato$^+$ cells. First, these cells may have recently class-switched to IgG3 and therefore still express surface IgM. However, sorted IgM$^+$Tomato$^+$ splenic B cells retained surface IgM after 48 hr in culture and did not gain expression of IgG3 (*Figure 1—figure supplement 2A*). Furthermore, single-cell RT-PCR of in vitro LPS stimulated Tomato$^-$ splenocytes revealed that the resulting IgM$^+$IgG3$^-$Tomato$^+$ B cells expressed *Ighm* mRNA but not *Ighg3* mRNA (*Figure 1—figure supplement 2B,D*; *Figure 1—figure supplement 3A*). Altogether, these results argue against the possibility that IgG3$^-$IgM$^+$Tomato$^+$ cells lack IgG3 because they recently class switched to IgG3. Second, we ruled out that *Ighg3*$^{T2A-Cre:TdTomato}$ mice were simply defective in IgG3 class switch recombination (CSR) by inducing IgG3 CSR via LPS stimulation of splenocytes in vitro (*Figure 1—figure supplement 2B,G*).

Therefore, we considered the possibility that IgG3$^-$IgM$^+$Tomato$^+$ cells express Cre without class switching to IgG3. The simplest way Cre expression could occur in the absence of CSR is translation of the T2A-Cre fusion protein from the *Ighg3* germ-line transcript (GLT), which precedes IgG3 CSR, especially since there is an in frame ATG upstream of the *Cre recombinase* gene (*Figure 1—figure supplement 2C*). Such a mechanism would not be unprecedented, as previous work by Wabl and colleagues showed the translatability of the *Ighm* GLT (*Bachl et al., 1996*). As predicted, IgM$^+$IgG3$^-$Tomato$^+$ B cells expressed both *Ighm* mRNA and the *Ighg3* GLT (*Figure 1—figure supplement 2E*; *Figure 1—figure supplement 3A*). Thus, the *Ighg3*$^{T2A-Cre:TdTomato}$ mouse appears to report induction of the *Ighg3* GLT rather than class switching to IgG3. Moreover, the presence of large numbers of IgG3$^-$IgM$^+$Tomato$^+$ cells indicates that a significant fraction of B cells has received signals that induce *Ighg3* GLT but not CSR to IgG3.

When we examined different subsets of B cells from *Ighg3*$^{T2A-Cre:TdTomato}$ mice in vivo, we discovered that the highest frequency of Tomato expression was in IgM$^+$ B-1a cells (CD19$^+$CD43$^+$CD5$^+$) (*Figure 1C*; *Figure 1—figure supplement 5C,F*). To begin to characterize the signals controlling

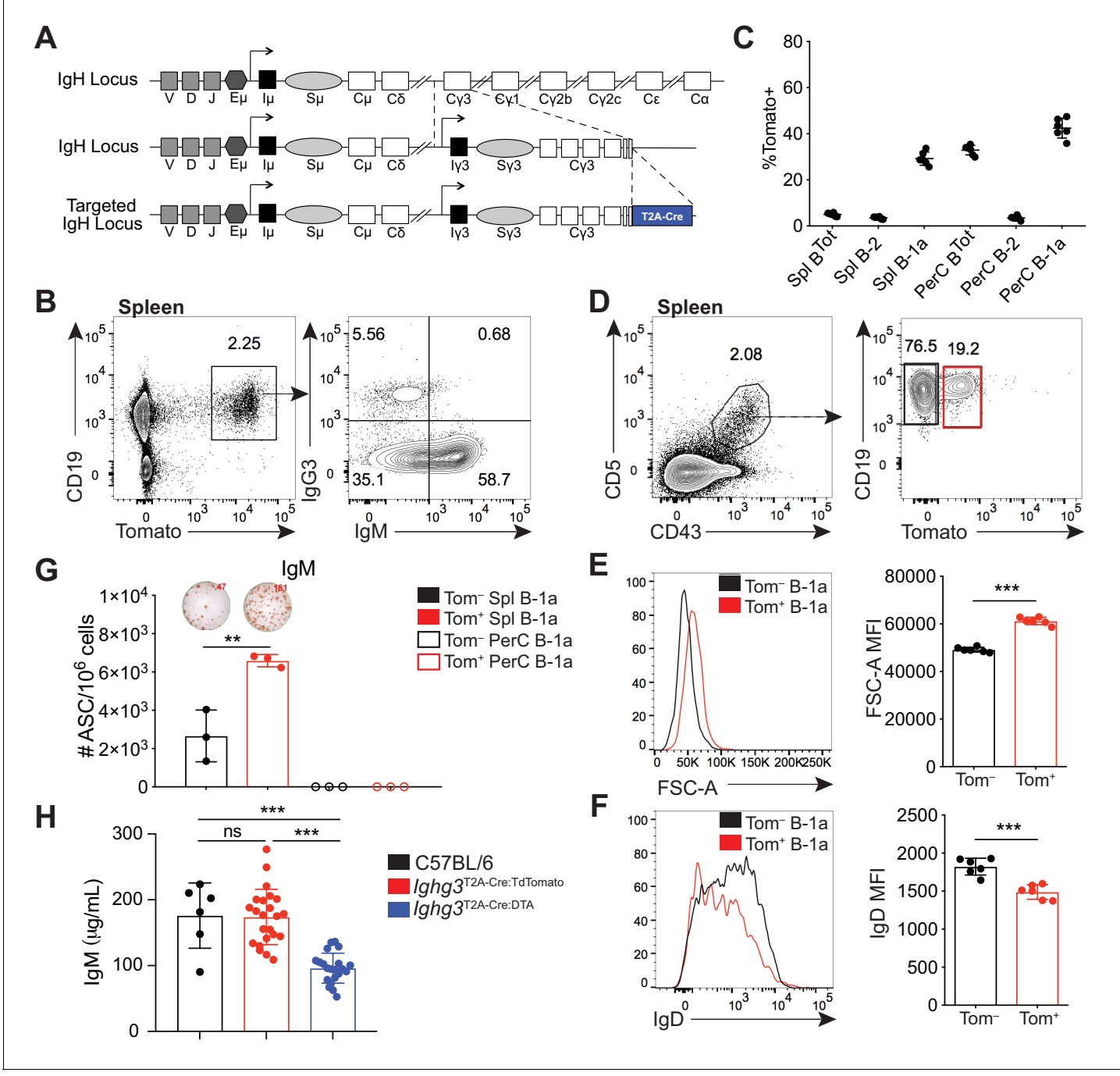

**Figure 1.** A reporter mouse marks activated B-1a cells. (A) Schematic of targeted insertion of *T2A-Cre* into the *Ighg3* (Iγ3) heavy chain locus to generate the *Ighg3*[T2A-Cre] reporter mouse. (B) Representative flow cytometry plot of IgG3 and IgM expression on pregated CD19+Tomato+ splenocytes from 6 wk old *Ighg3*[T2A-Cre] mice crossed to *Rosa26*[STOP-flox-TdTomato] mice (*Ighg3*[T2A-Cre:TdTomato]). (C) Percentage Tomato expression in B cell subsets from the peritoneal cavity (PerC) and spleen (Spl) of 6 wk old *Ighg3*[T2A-Cre:TdTomato] mice by flow cytometry. Total B cells defined as CD19+; B-2 cells defined as CD19+CD23+CD43−CD5−; B-1a cells defined as CD19+CD23−CD43+CD5+. (D) Representative flow cytometry gating of Tomato− and Tomato+ splenic B-1a cells from 3 wk old *Ighg3*[T2A-Cre:TdTomato] mice, pregated on Live/CD19+CD23− cells. (E) Representative histogram and quantification of FSC-A and (F) IgD expression on pre-gated Tomato− (black) and Tomato+ (red) splenic B-1a cells. (G) Number of IgM+ antibody secreting cells (ASCs)/10^6 present in purified Tomato− (black) or Tomato+ (red) splenic (closed circles) or peritoneal cavity (PerC) (open circles) B-1a cells from 6 wk old *Ighg3*[T2A-Cre:TdTomato] mice, as measured by ELISpot. (H) Serum IgM titers of 7 wk old C57BL/6 (black), *Ighg3*[T2A-Cre:TdTomato] (red) and *Ighg3*[T2A-Cre] mice crossed to *Rosa26*[STOP-flox-DTA] (*Ighg3*[T2A-Cre:DTA]) (blue) mice. Error bars indicate the mean (± SEM). *p<0.05, **p<0.01, and ***p<0.001 (unpaired two-tailed Student's t-test). Each data point represents an individual mouse (C, E-H). Data are representative of at least three independent experiments (B-G) or pooled from five independent experiments (H).

*Figure 1 continued on next page*

*Figure 1 continued*

DOI: https://doi.org/10.7554/eLife.47015.002

The following figure supplements are available for figure 1:

**Figure supplement 1.** Validation of correct targeting of knock-in mouse.

DOI: https://doi.org/10.7554/eLife.47015.003

**Figure supplement 2.** TdTomato expression in IgM+ cells correlates with Ighg3 germ-line transcription in Ighg3 reporter mouse.

DOI: https://doi.org/10.7554/eLife.47015.004

**Figure supplement 3.** Single-cell RT-PCR Primers.

DOI: https://doi.org/10.7554/eLife.47015.005

**Figure supplement 4.** B cell development in bone marrow is unaltered in reporter mouse.

DOI: https://doi.org/10.7554/eLife.47015.006

**Figure supplement 5.** Characterization of B cell subsets in the spleen and peritoneal cavity of reporter mice.

DOI: https://doi.org/10.7554/eLife.47015.007

induction of Cre in B cells of $Ighg3^{T2A-Cre:TdTomato}$ mice, we tracked Tomato expression in B cells in vivo. The Tomato$^+$ splenic B-1a cells were larger in size (*Figure 1D–E*) and had lower surface IgD expression (*Figure 1F*) relative to Tomato$^-$ splenic B-1a cells. Moreover, the frequency of cells spontaneously secreting IgM was much higher in sorted Tomato$^+$ B-1a cells when compared to Tomato$^-$ B-1a cells (*Figure 1G*). Based on these findings, we considered the possibility that the $Ighg3^{T2A-Cre:TdTomato}$ mouse marks a subset of activated B-1a cells and that activation of these cells correlates with induction of the Ighg3 GLT rather than CSR to IgG3. To test this model, we stimulated splenocytes from $Ighg3^{T2A-Cre:TdTomato}$ mice with LPS in vitro. Induction of Tomato expression correlated with reduction of surface IgD expression and increased size, indicative of cells that have undergone activation and/or plasmablast differentiation (*Figure 1—figure supplement 2G*). Finally, to test the functional significance of the Cre-expressing cells in $Ighg3^{T2A-Cre}$ mice, we crossed $Ighg3^{T2A-Cre}$ mice with $Rosa26^{STOP-flox-DTA}$ mice to ablate any Cre-expressing cells due to forced expression of diphtheria toxin and induction of cell death. As expected, the resulting $Ighg3^{T2A-Cre:DTA}$ mice had a complete loss of serum IgG3 (*Figure 1—figure supplement 2F*), but also significantly reduced titers of serum IgM (*Figure 1H*), indicating that the Tomato$^+$ activated B-1a cells marked in the $Ighg3^{T2A-Cre:TdTomato}$ mice are a significant source of serum IgM.

Importantly, $Ighg3^{T2A-Cre:TdTomato}$ and $Ighg3^{T2A-Cre:DTA}$ mice had no apparent defects in B cell development in the bone marrow (*Figure 1—figure supplement 4A-C*). Moreover, Tomato expression was restricted to both mature and likely re-circulating B-1 cells in the bone marrow (*Figure 1—figure supplement 4D-E*). $Ighg3^{T2A-Cre:TdTomato}$ and $Ighg3^{T2A-Cre:DTA}$ mice also had normal absolute counts of follicular, marginal zone, B-1a, and B-1b cells in the spleen and peritoneal cavity (*Figure 1—figure supplement 5A,B,E*). Intriguingly, a significant fraction of Tomato$^+$CD19$^+$ cells in the spleens of $Ighg3^{T2A-Cre:TdTomato}$ mice lack IgM expression (*Figure 1B*, *Figure 1—figure supplement 5D*). However, the majority of these cells express the B-1 marker CD43 and likely represent another subset of spontaneously secreting B-1 cells recently described by Savage et. al. (*Figure 1—figure supplement 5D*) (*Savage et al., 2017*).

Thus, while the $Ighg3^{T2A-Cre:TdTomato}$ mouse did not function as we originally intended, our results indicate that the mouse can be used as a tool to track an activated, antibody-secreting subset of B-1a cells. This serendipitous outcome enabled us to study the signals that regulate B-1a cell activation in vivo.

## B-1a cells have a history of B cell receptor-mediated activation

B-1a cells are known to spontaneously secrete IgM, but the contribution of BCR-mediated antigenic stimulation to steady-state B-1a antibody secretion is not well understood. Because we determined that the $Ighg3^{T2A-Cre:TdTomato}$ mouse marks a subset of activated B-1a cells with enhanced 'spontaneous' IgM secretion, we next sought to determine the contribution of BCR signaling to Tomato expression. In agreement with recent studies, we found that B-1a cells express high levels of Nur77, a gene whose expression is tightly linked to BCR-mediated signaling (*Figure 2A*). Interestingly, we found that splenic B-1a cells express even higher levels of Nur77 than peritoneal cavity B-1a cells (*Figure 2A*), consistent with our findings and previous studies showing that splenic, but not peritoneal cavity, B-1a cells secrete IgM when cultured (*Figure 1G*), and that the spleen is the site where

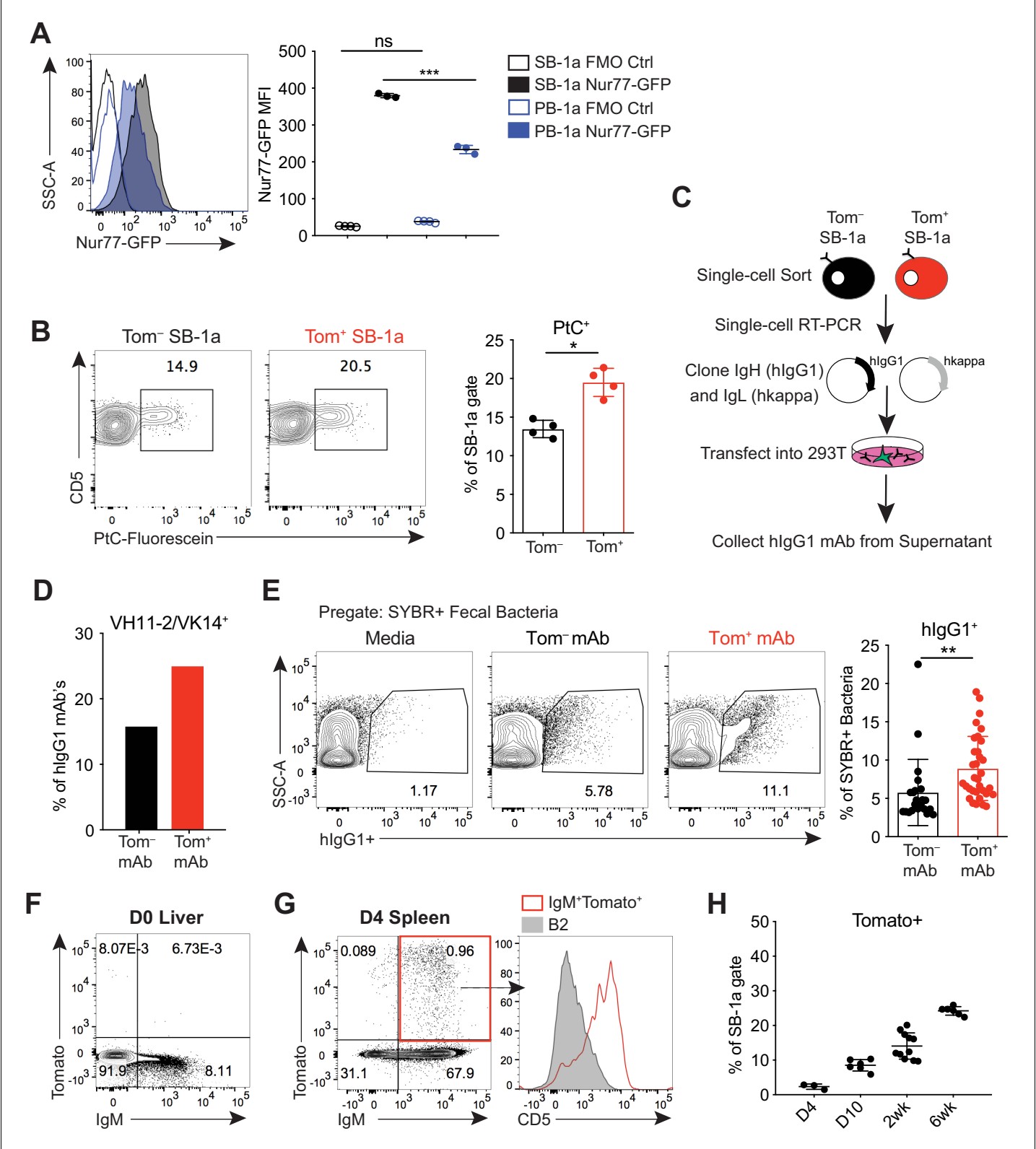

**Figure 2.** B-1a cells have a history of B cell receptor-mediated activation. (A) Representative flow cytometry histogram (left) and quantification of mean fluorescence intensity (right) of GFP expression on splenic (SB-1a) (black) and peritoneal cavity (PB-1a) (blue) B-1a cells (CD19⁺CD23⁻CD43⁺CD5⁺) in 3 wk old Nur77-GFP reporter mice (filled histogram, closed circles) or reporter-negative mice as controls (FMO Ctrl) (unfilled histogram, open circles). (B) Representative flow cytometry plots (left) and quantification (right) of the percentage of Tomato⁻ and Tomato⁺ splenic B-1a cells from 6 wk old

*Figure 2 continued on next page*

*Figure 2 continued*

*Ighg3*[T2A-Cre:TdTomato] mice stained with fluorescein-labeled phosphatidylcholine liposomes. (**C**) Schematic showing the generation of hIgG1 recombinant monoclonal antibodies (mAb) from single-cell sorted Tomato⁻ or Tomato⁺ splenic B-1a cells (SB-1a) from 6wk old *Ighg3*[T2A-Cre:TdTomato] mice. There are two source files associated with this figure with a comprehensive description of all of the mAbs generated. (**D**) The percentage VH11-2/VK14 gene usage in monoclonal antibodies (mAbs) generated from Tomato⁻ (black) and Tomato⁺ (red) splenic B-1a (SB-1a) cells from 6 wk old *Ighg3*[T2A-Cre:TdTomato] mice, as described in (C); Tom⁺ mAb n= 48; Tom⁻ mAb n= 37. (**E**) Representative flow cyometry plots (**left**) and quantification (**right**) of pregated SYBR⁺ fecal bacteria bound by recombinant hIgG1 monoclonal antibodies generated as described in (C); PtC-reactive VH11-2/VK14 expressing mAb's were excluded from this analysis; Tom⁺ mAb n= 32; Tom⁻ mAb n= 23. (**F**) Representative flow cytometry plot of IgM versus Tomato expression in pre-gated CD19⁺ D0 liver cells or (**G**) CD19⁺ D4 spleen cells from *Ighg3*[T2A-Cre:TdTomato] mice; representative flow cytometry histogram of CD5 expression on pre-gated CD19⁺IgM⁺Tomato⁺ D4 spleen cells (red, unfilled) compared to CD19⁺IgM⁺CD23⁺CD43⁻CD5⁻ B2 cells (gray, filled) (G, right). (**H**) Quantification of percentage Tomato expression in B-1a cells from the spleens (SB-1a) of D4, D10, 2 wk, and 6 wk old *Ighg3*[T2A-Cre:TdTomato] mice by flow cytometry. Error bars indicate the mean (± SEM). *p<0.05, **p<0.01, and ***p<0.001 (unpaired two-tailed Student's t-test (B, E) or one-way ANOVA (A)). Each data point represents an individual mouse (A,B,H) or monoclonal antibody (E). Data are representative of at least three independent experiments (A-B, E-H).

DOI: https://doi.org/10.7554/eLife.47015.008

The following source data is available for figure 2:

**Source data 1.** Summary of monoclonal antibodies generated from Tomato⁻ spleen B-1a cells.
DOI: https://doi.org/10.7554/eLife.47015.009

**Source data 2.** Summary of monoclonal antibodies generated from Tomato⁺ spleen B-1a cells.
DOI: https://doi.org/10.7554/eLife.47015.010

B-1a cells are initially activated (*Wardemann et al., 2002*). We therefore focused our analysis on splenic B-1a cells.

To explore more directly whether B-1a activation and antibody secretion are mediated by BCR signaling, we probed the BCR specificities of splenic Tomato⁺ B-1a cells to determine whether certain specificities were enriched relative to Tomato⁻ B-1a cells. Using fluorescently labeled PtC-coated liposomes to quantify PtC-reactive B-1a cells by flow cytometry, we observed a higher frequency of PtC-reactivity within Tomato⁺ B-1a cells versus Tomato⁻ cells (*Figure 2B*). We also cloned the immunoglobulin light (IgL) and heavy (IgH) chain genes from individual Tomato⁺ and Tomato⁻ splenic IgM⁺ B-1a cells from adult mice and generated recombinant monoclonal antibodies (mAbs) using a hIgG1 scaffold (*Tiller et al., 2009*) (*Figure 2C*). Consistent with the higher frequency of PtC-reactive Tomato⁺ splenic B-1a cells measured by flow cytometry, the canonical VH11-2/VK14 heavy and light chain paired genes that encode PtC-reactive BCRs were enriched in mAbs generated from Tomato⁺ B-1a cells in comparison to mAbs generated from Tomato⁻ B-1a cells (*Figure 2D*).

We also examined the reactivity of mAbs generated from Tomato⁺ and Tomato⁻ B-1a cells with the microbiota. We excluded the PtC-reactive antibodies from this analysis to ensure that any observed skewing was not due to this specificity. Binding of each mAb to fecal microbiota was measured by flow cytometry, using an assay we have previously described (*Koch et al., 2016*). Briefly, after incubating with individual hIgG1 mAbs, fecal contents are stained with the DNA-binding dye SYBR to differentiate bacteria (SYBR⁺) from other components present in the feces such as food (SYBR⁻). Remarkably, mAbs generated from Tomato⁺ B-1a cells were significantly more reactive (*Figure 2E*). While this assay cannot track specific epitopes, the data indicate that the BCRs of Tomato⁺ cells are enriched for specificities against the microbiota.

Altogether, the increased frequency of BCR specificities for common B-1a antigens like PtC and the microbiota in the activated, antibody-secreting Tomato⁺ pool of splenic B-1a cells indicates that antigen-specific BCR signaling, as opposed to antigen-independent polyclonal expansion, mediates the activation and function of at least certain B-1a cells at steady state. In further support of this conclusion, we did not detect Tomato expression in D0 neonatal liver IgM⁺CD19⁺ B cells (*Figure 2F*); instead, Tomato⁺ cells first appeared at about 4 days of age in splenic CD5⁺ B-1a cells (*Figure 2G*) and increased in frequency during the first few weeks of life (*Figure 2H*). Thus, Tomato expression correlates with the acquisition of a microbiota and migration of B-1a cells from the neonatal liver to distal sites such as the spleen, where they likely first encounter antigen.

## Colonization with a microbiota is required for the production of B-1 derived microbiota-reactive IgM

We next sought to determine the extent to which self antigens versus microbial antigens impact B-1a responses. A recent study reported no significant difference in B-1a immunoglobulin heavy chain repertoire between adult GF and SPF mice, favoring a model whereby microbiota-reactive serum IgM present in healthy SPF mice arises from antibodies selected on self antigens that are cross-reactive to microbial antigens present on constituents of the microbiota (e.g. PtC) (*Yang et al., 2015*). However, our results described in the previous section raise the possibility that other microbiota-derived epitopes can trigger antibody production by B-1a cells with distinct specificities.

Before investigating the importance of various antigens for driving the expansion and/or secretion of microbiota-reactive B-1a cells, we first determined what fraction of the microbiota-reactive IgM is derived from B-1 cells by generating Ig allotype disparate chimeric mice. In this system IgH$^b$ peritoneal cavity B-1 cells are transferred into neonatal mice of the IgH$^a$ allotype that are treated with anti-IgM$^a$ for the first 6 weeks of life. This treatment depletes the host IgH$^a$ B cells but not the transferred IgM$^b$ B-1 cells. The anti-IgM$^a$ treatment is stopped at 6 weeks of age, at which point endogenous B-1 cells are no longer generated, since they are of fetal and neonatal origin (*Lalor et al., 1989b*). However, bone marrow-derived IgH$^a$ cells are able to reconstitute the B-2 cell compartment, allowing one to determine the cellular origin of antibodies using Ig allotype specific antibodies (*Figure 3A*). Consistent with previous studies validating the B-1 chimera experimental model (*Lalor et al., 1989a*; *Savage et al., 2017*; *Baumgarth et al., 1999*), we found that the majority of serum IgM in adult B-1 chimeras was B-1 derived. This was determined by the almost complete absence of endogenous B-2 derived serum IgM$^a$ (also serving as a control for efficient endogenous B-1 cell depletion) (*Figure 3B*). Moreover, in adult B-1 chimeric mice essentially all microbiota-reactive IgM was derived from IgM$^b$ B-1 cells, comparable to the levels of microbiota-reactive IgM in WT mice (*Figure 3C,E*). Thus, B-1 cells are the main producers of microbiota-reactive IgM.

Next we asked whether colonization with a microbiota is necessary for B-1a cells to produce microbiota-reactive IgM. Staining of fecal bacteria from $\mu MT^{-/-}$ mice with sera from SPF or GF mice revealed that GF mice produce significantly reduced titers of microbiota-reactive IgM (*Figure 3D–E*), despite normal serum IgM titers (*Figure 3F*). In contrast, the frequency of PtC-reactive B-1a cells in the peritoneal cavity and spleen was similar in SPF and GF mice (*Figure 3G–H*), consistent with previous reports (*Hooijkaas et al., 1984*; *Bos et al., 1989*; *Haury et al., 1997*). These data suggest that steady state microbiota-reactive IgM cannot merely be explained by the cross-reactivity of antibodies produced by B-1a cells in response to self-antigens; instead, microbiota-reactive antibody production by B-1a cells is dependent on microbial colonization. Importantly, these results also demonstrate different requirements for the production of microbiota-reactive versus PtC-reactive IgM.

## Loss of Toll-like receptor signaling results in reduced B-1a responses to both phosphatidylocholine and the microbiota

Our results thus far provide evidence that B-1a cells require BCR signaling for their selection and activation, yet previous work from numerous groups have suggested that B-1a cells are non-responsive to BCR cross-linking and instead respond in a non-clonal fashion to TLR ligands (*Ha et al., 2006*; *Genestier et al., 2007*). Indeed, TLR ligands induce B-1a cell proliferation, plasma cell differentiation, and CSR in vitro, whereas IgM crosslinking induces apoptosis (*Morris and Rothstein, 1993*; *Bikah et al., 1996*; *Ochi and Watanabe, 2000*). Moreover, with the use of TLR reporter mice generated in our lab (*Price et al., 2018*; *Roberts et al., 2017*), we determined that B-1a cells express significantly higher levels of TLR2, TLR4, TLR7, and TLR9 than B-2 cells (*Figure 4—figure supplement 1*). Whether TLRs are required for B-1a function in vivo and how TLR signaling may integrate with BCR signaling to control B-1a activation are unanswered questions.

We used mice deficient in TLR signaling to assess the contributions of TLRs to B-1a function. Specifically, we analyzed $Tlr2^{-/-}Tlr4^{-/-}Unc93b1^{3d/3d}$ mice, which lack TLR2 and TLR4 and have a mutation in $Unc93b1$, which renders all remaining TLRs non-functional. Unlike $Myd88^{-/-}Trif^{-/-}$ mice, $Tlr2^{-/-}Tlr4^{-/-}Unc93b1^{3d/3d}$ mice have functional IL-1 receptor family signaling and do not suffer from intestinal barrier defects which may complicate analysis (*Sivick et al., 2014*). Additionally, $Tlr2^{-/-}Tlr4^{-/-}Unc93b1^{3d/3d}$ mice allow us to dissect the contribution of subsets of TLRs on B-1a antibody production (i.e. $Tlr2^{-/-}Tlr4^{-/-}$ vs. $Unc93b1^{3d/3d}$). $Tlr2^{-/-}Tlr4^{-/-}Unc93b1^{3d/3d}$ mice had the same

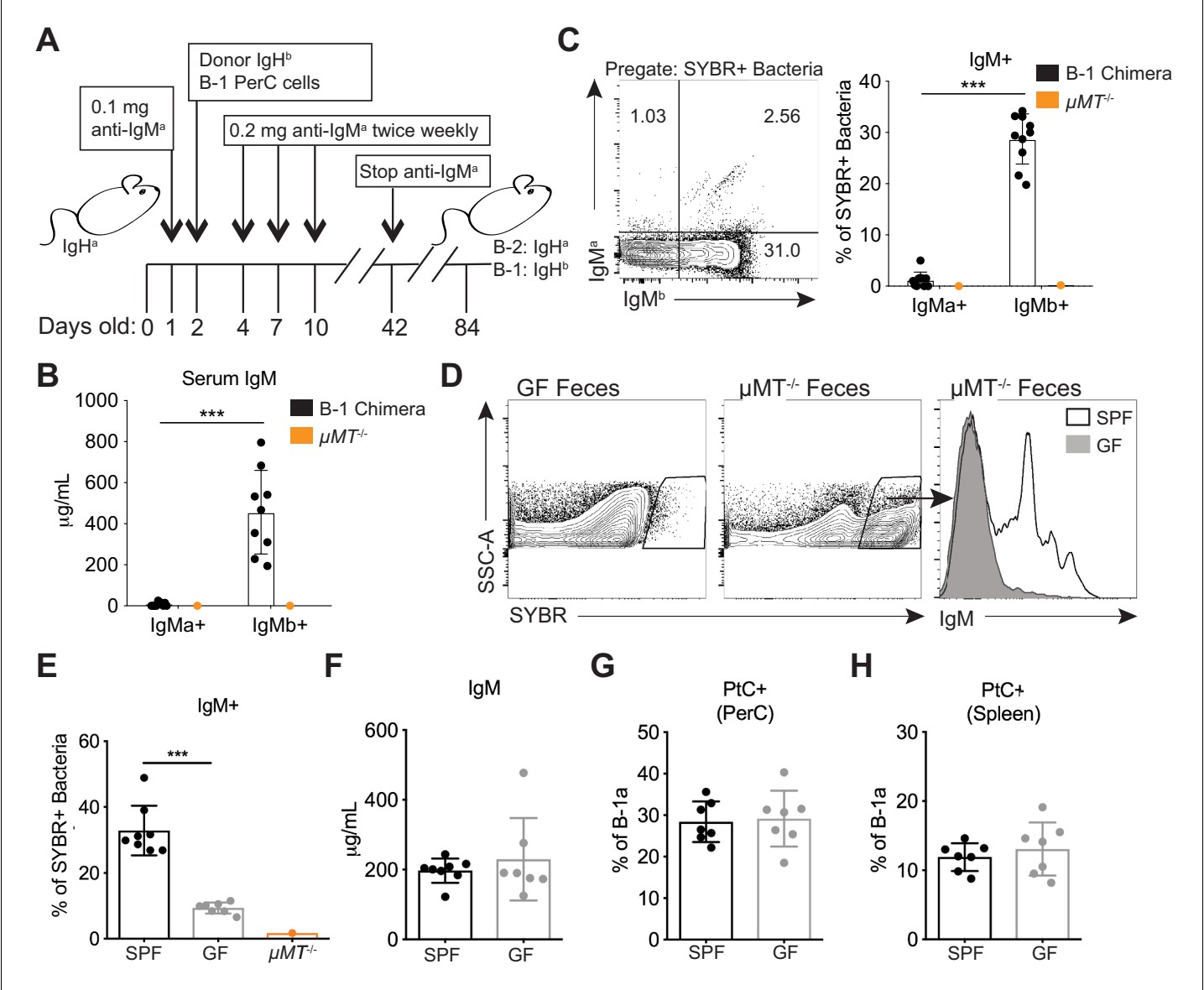

**Figure 3.** Colonization of a microbiota is required for B-1 derived microbiota-reactive IgM. (A) Schematic showing the generation of B-1 disparate allotype chimera mice. (B) Serum titers of B-2-derived IgMa and B-1-derived IgMb in 14 wk old B-1 chimeras, as described in (A). (C) Representative flow cytometry plot (left) and quantification (right) of percentage of SYBR$^+$ fecal bacteria bound by B-1-derived serum IgMb and B-2-derived serum IgMa from 14 wk old B-1 chimeras, as described in (A); $\mu MT^{-/-}$ mouse serum included as a negative control (orange). (D) Representative SYBR staining of germ-free (GF) (left) or SPF $\mu MT^{-/-}$ (middle) feces, as measured by flow cytometry. Representative flow cytometry histogram plot showing IgM staining of SYBR$^+\mu MT^{-/-}$ fecal bacteria with serum from 7 wk old SPF (black line, no fill) or germ free (GF) mice (gray line, filled) (right). (E) Quantification of percentage of SYBR$^+\mu MT^{-/-}$ fecal bacteria bound by serum IgM from seven wk old SPF (black), GF (gray), or $\mu MT^{-/-}$ control (orange) mice (right). (F) Serum titers of IgM in 7 wk old SPF (black) or GF (gray) mice, as measured by ELISA. (G-H) Percentage fluorescein-labeled PtC-liposome positive (PtC$^+$) B-1a cells (CD19$^+$CD23$^-$CD43$^+$CD5$^+$) in the (G) peritoneal cavity (PerC) and (H) spleen of 7 wk old SPF (black) or GF (gray) mice, as measured by flow cytometry. Each data point represents an individual mouse (B-C, E-H). Error bars indicate the mean (± SEM). *p<0.05, **p<0.01, and ***p<0.001 (unpaired two-tailed Student's t-test). Data are representative of at least three independent experiments (B-H).

DOI: https://doi.org/10.7554/eLife.47015.011

frequency of B-1a cells in the peritoneal cavity (*Figure 4A*) and the spleen (*Figure 4B*), suggesting that TLR signaling is not required for general B-1a cell development. *Tlr2$^{-/-}$Tlr4$^{-/-}$Unc93b1$^{3d/3d}$* mice had and comparable IgM titers to WT mice (*Figure 4C*), suggesting that TLR signaling is not reguired for general B-1a antibody secretion, although we noted a slight reduction that was not

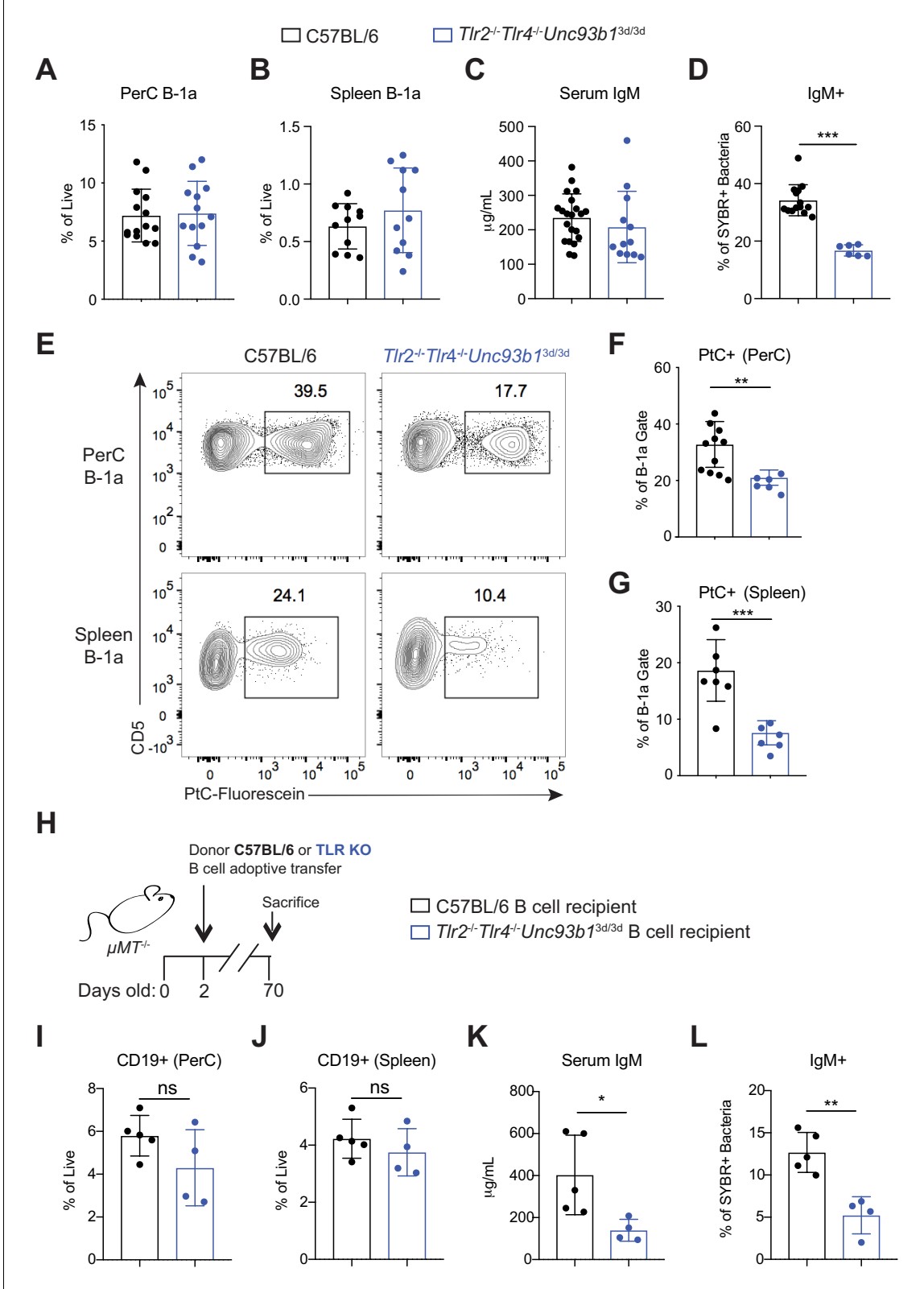

**Figure 4.** Loss of Toll-like receptor signaling results in reduced B-1a responses to both phosphatidylcholine and the microbiota. (A) Frequency of live B-1a cells in 7 wk old WT (black) or *Tlr2⁻/⁻Tlr4⁻/⁻Unc93b1*³ᵈ/³ᵈ (blue) mice in the peritoneal cavity (PerC) and (B) spleen, as measured by flow cytometry. B-1a cells defined as CD19⁺CD23⁻CD43⁺CD5⁺. (C) Serum IgM titers in 7 wk old WT (black) or *Tlr2⁻/⁻Tlr4⁻/⁻Unc93b1*³ᵈ/³ᵈ (blue) mice, as measured by ELISA. (D) Percentage of SYBR⁺ *μMT⁻/⁻* fecal bacteria bound by serum IgM from WT (black) or *Tlr2⁻/⁻Tlr4⁻/⁻Unc93b1*³ᵈ/³ᵈ (blue) mice, as measured by flow
*Figure 4 continued on next page*

Figure 4 continued

cytometry. (E) Representative flow cytometry plot and quantification (F-G) of percentage of B-1a cells in the (F) peritoneal cavity (PerC) and (G) spleen bound by fluorescein-labeled phosphatidylcholine liposomes in 7 wk old WT (black) or $Tlr2^{-/-}Tlr4^{-/-}Unc93b1^{3d/3d}$ (blue) mice. (H) Schematic showing adoptive transfer of CD19$^+$ pooled bone marrow and spleen cells (B cells) from either C57BL/6 (WT) (black) or $Tlr2^{-/-}Tlr4^{-/-}Unc93b1^{3d/3d}$ (blue) into B cell deficient $\mu MT^{-/-}$ recipeint mice at 2 days of age and sacrificed at 10 weeks of age. (I-J) Frequency of live B-1a cells in 7 wk old WT (black) or $Tlr2^{-/-}Tlr4^{-/-}Unc93b1^{3d/3d}$ (blue) B cell recipient mice in the (J) peritoneal cavity (PerC) and (J) spleen, as measured by flow cytometry. (K) Serum IgM titers in 7 wk old WT (black) or $Tlr2^{-/-}Tlr4^{-/-}Unc93b1^{3d/3d}$ (blue) B cell recipient mice, as measured by ELISA. (L) Percentage of SYBR$^+$ fecal bacteria bound by serum IgM from WT (black) or $Tlr2^{-/-}Tlr4^{-/-}Unc93b1^{3d/3d}$ (blue) B cell recipient mice, as measured by flow cytometry. Error bars indicate the mean (± SEM). *p<0.05, **p<0.01, and ***p<0.001 (unpaired two-tailed Student's t-test). Each data point represents an individual mouse. Data are pooled from at least three independent experiments (A-G; I-L).

DOI: https://doi.org/10.7554/eLife.47015.012

The following figure supplements are available for figure 4:

Figure supplement 1. TLR expression on B cell subsets using TLR reporter mice.
DOI: https://doi.org/10.7554/eLife.47015.013

Figure supplement 2. Loss of Toll-like receptor signaling results in reduced B-1a responses to both phosphatidylcholine and the microbiota in co-housed mice.
DOI: https://doi.org/10.7554/eLife.47015.014

statistically significant. However, we observed a significant reduction in microbiota-reactive IgM in $Tlr2^{-/-}Tlr4^{-/-}Unc93b1^{3d/3d}$ mice (Figure 4D), and a marked reduction in the frequency of PtC-reactive B-1a cells in both the peritoneal cavity and spleen (Figure 4E–G). Importantly, these TLR-dependent B-1a responses were also observed in WT and $Tlr2^{-/-}Tlr4^{-/-}Unc93b1^{3d/3d}$ mice after co-housing, showing that the significant reduction in PtC- and microbiota- reactive B-1a responses in $Tlr2^{-/-}Tlr4^{-/-}Unc93b1^{3d/3d}$ mice are not due to differences in the microbiota between separately housed strains of mice (Figure 4—figure supplement 2). When considered together with our earlier results, these data indicate that TLR and BCR signaling are both required for activation of distinct subsets of B-1a cells. Not all B-1a cells are subject to this dual requirement, though, as total frequencies of B-1a cells and total serum IgM titers appeared normal in $Tlr2^{-/-}Tlr4^{-/-}Unc93b1^{3d/3d}$ mice.

In order to determine whether the requirement for TLRs was B-cell intrinsic, we performed adoptive transfers of B cells from WT or $Tlr2^{-/-}Tlr4^{-/-}Unc93b1^{3d/3d}$ mice into B-cell deficient $\mu MT^{-/-}$ neonates and measured serum IgM responses 10 weeks post transfer (Figure 4H). Despite comparable frequencies of donor CD19$^+$ cells in the PerC and spleen of adult recipient mice (Figure 4I–J), we observed significantly lower total serum IgM titers (Figure 4K) and a concomitant decrease in microbiota reactivity (Figure 4L) in mice adoptively transferred with $Tlr2^{-/-}Tlr4^{-/-}Unc93b1^{3d/3d}$ B cells. Of note, we did not see this same reduction in serum IgM titers in global $Tlr2^{-/-}Tlr4^{-/-}Unc93b1^{3d/3d}$ mice (Figure 4C). It is therefore possible that the apparently normal serum IgM titers that we observe in global $Tlr2^{-/-}Tlr4^{-/-}Unc93b1^{3d/3d}$ mice is caused by a compensatory increase in serum IgM due to TLR deficiency in additional cells types. Altogether, these data show that B-cell cell intrinsic TLR signaling is required for both antibody secretion as well as microbiota reactivity.

## TLR2 and TLR4 regulate microbiota-reactive B1a responses, whereas Unc93b1-dependent TLRs regulate anti-PtC reactivities

To further investigate how TLRs control activation of microbiota-reactive and PtC-reactive B-1a cells, we examined mice lacking subsets of TLRs. We broadly divided TLRs into two categories: TLRs involved in recognition of pathogen associated molecular patterns (PAMPs) on the surface of microbes (TLR2 and TLR4) and TLRs that localize to endosomes and recognize self nucleic acids and PAMPs within microbes (TLR3, TLR7, TLR9, TLR11, TLR12, and TLR13). All TLRs in the latter category require Unc93B1 to function, enabling us to use $Unc93b1^{3d/3d}$ mice to study the importance of these TLRs. Note that TLR5 is also Unc93b1-dependent but is not expressed on B cells (Figure 4—figure supplement 1). Both $Tlr2^{-/-}Tlr4^{-/-}$ and $Unc93b1^{3d/3d}$ mice had normal frequencies of B-1a cells in the peritoneal cavity (Figure 5A) and spleen (Figure 5B) as well as normal serum IgM titers (Figure 5C), consistent with our previous results analyzing global TLR deficient mice (Figure 4). Strikingly, mice lacking function of specific subsets of TLRs showed differential defects in B-1a responses. $Unc93b1^{3d/3d}$ mice had normal levels of microbiota-reactive IgM (Figure 5D) but significantly reduced frequencies of PtC-reactive B-1a cells in both the peritoneal cavity (Figure 5E) and the

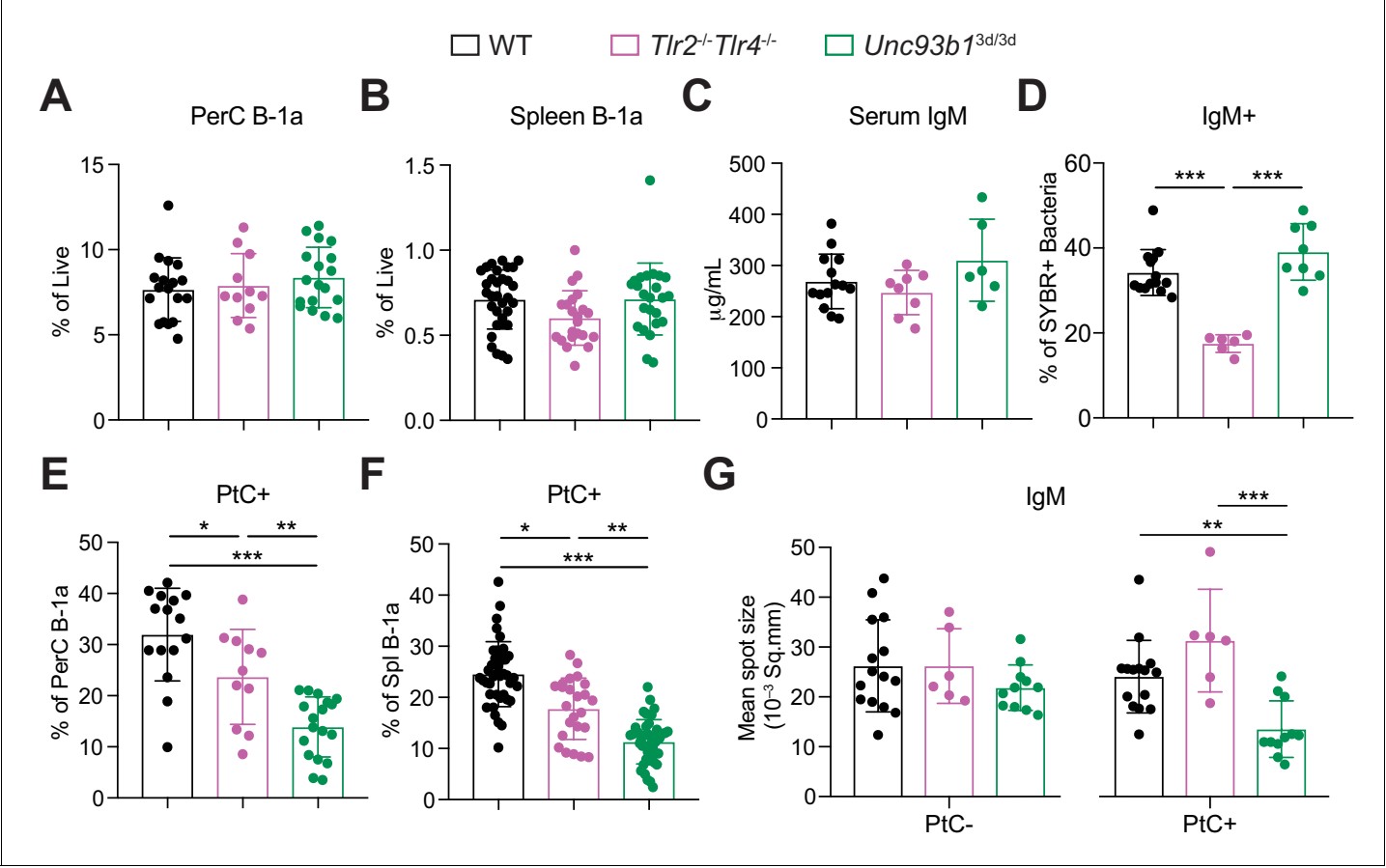

**Figure 5.** TLR2 and TLR4 regulate B-1a responses to the microbiota, whereas Unc93B1-dependent TLRs regulate phosphatidylcholine-reactive B-1a responses. (A–B) Frequency of live B-1a cells in 7 wk old WT (black), *Tlr2*−/−*Tlr4*−/− (pink), or *Unc93b1*3d/3d (green) mice in the (A) peritoneal cavity (PerC) and (B) spleen, as measured by flow cytometry. (C) Serum IgM titers in 7 wk old WT (black), *Tlr2*−/−*Tlr4*−/− (pink), or *Unc93b1*3d/3d (green) mice, as measured by ELISA. (D) Percentage of SYBR+*μMT*−/− fecal bacteria bound by serum IgM from 7 wk old WT (black), *Tlr2*−/−*Tlr4*−/− (pink), or *Unc93b1*3d/3d (green) mice, as measured by flow cytometry. (E-F) Percentage fluorescein-labeled phosphatidylcholine-liposome positive (PtC+) (E) peritoneal cavity (PerC) B-1a and (F) spleen (Spl) B-1a cells in 7 wk old WT (black), *Tlr2*−/−*Tlr4*−/− (pink), or *Unc93b1*3d/3d (green) mice, as measured by flow cytometry. (G) Quantification of the mean spot size of IgM secreted by sorted PtC– (left) or PtC+ (right) splenic B-1a cells from WT (black), *Tlr2*−/−*Tlr4*−/− (pink), or *Unc93b1*3d/3d (green) mice after 20 hr in culture, as measured by ELISpot. Error bars indicate the mean (± SEM). *p<0.05, **p<0.01, and ***p<0.001 (One-way ANOVA). Each data point represents an individual mouse. Data are pooled from at least three independent experiments (A-G).

DOI: https://doi.org/10.7554/eLife.47015.015

spleen (*Figure 5F*). In contrast, *Tlr2*−/−*Tlr4*−/− mice had a significant reduction in microbiota-reactive IgM (*Figure 5D*) but only slightly reduced frequencies of PtC-reactive B-1a cells (*Figure 5E–F*). Furthermore, IgM ELISpot analysis of PtC-reactive splenic B-1a cells from WT, *Tlr2*−/−*Tlr4*−/−, and *Unc93b1*3d/3d mice revealed that Unc93B1-dependent TLRs, but not TLR2 and TLR4, are required for optimal secretion of PtC-reactive IgM (*Figure 5G*).

Because there are currently no tools to track microbiota-reactive B-1a cells in vivo, we were unable to directly correlate the reduction in microbiota-reactive serum IgM in mice lacking TLR2 and TLR4 with changes in the B-1a repertoire. Therefore, in order to investigate the role of specific subsets of TLRs in regulating the B-1a repertoire more directly, we sorted B-1a cells from the spleen and peritoneal cavity of WT, *Tlr2*−/−*Tlr4*−/−*Unc93b1*3d/3d, *Unc93b1*3d/3d, and *Tlr2*−/−*Tlr4*−/− mice (*Figure 6A*), and performed bulk RNA immunoglobulin heavy chain sequencing. Overall, the VH gene usage in WT, *Tlr2*−/−*Tlr4*−/−*Unc93b1*3d/3d, *Unc93b1*3d/3d, and *Tlr2*−/−*Tlr4*−/− mice was almost indistinguishable (*Figure 6—figure supplement 1*). However, we did observe a significant reduction in the frequency of VH9-3 and VH3-6 gene usage in B-1a samples from mice lacking TLR2 and TLR4 that was not present in mice lacking only Unc93B1 (*Figure 6B–C*). Intriguingly, we had previously

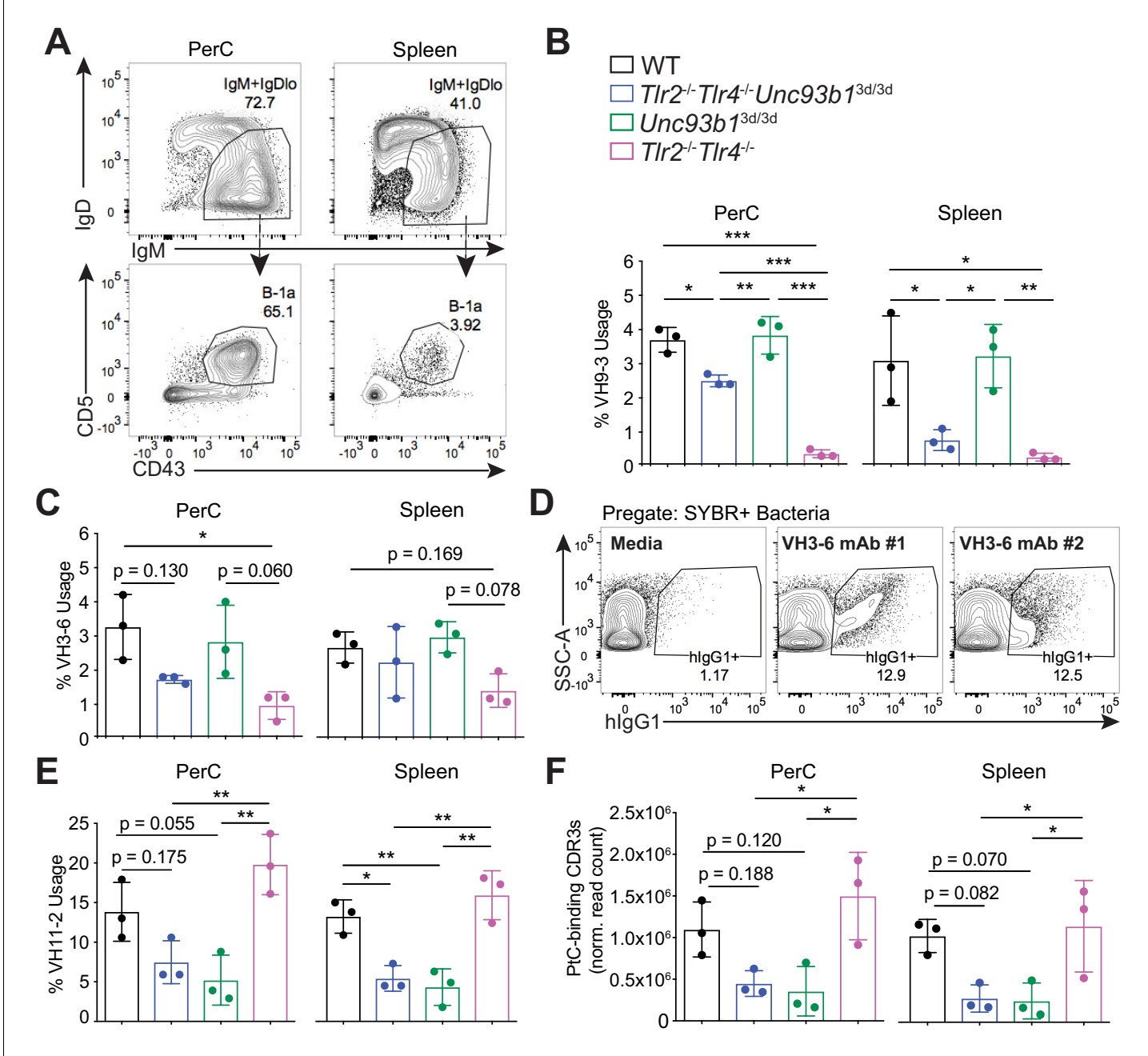

**Figure 6.** B-1a immunoglobulin repertoire analysis reveals unique regulation of a subset of heavy chain genes by distinct subsets of TLRs. (A) Representative flow cytometry gating strategy for sorting IgM⁺IgD^lo CD43⁺CD5⁺B-1a cells (pregated as CD19⁺/DAPI⁻) in the peritoneal cavity (PerC) (left) and spleen (right). Prior to sorting, splenocytes were depleted of CD3⁺CD4⁺CD8⁺F4/80⁺NK1.1⁺GR-1⁺ cells using biotinylated antibodies and streptavidin magnetic beads. (B-C) The percentage of heavy chain CDR3 nucleotide sequencing reads expressing the germline (B) VH9-3 allele or the (C) VH3-6 allele in peritoneal cavity (PerC) (left) and spleen (right) B-1a samples (% usage). (D) Percentage of pre-gated SYBR⁺ fecal bacteria bound by two individual recombinant hIgG1 monoclonal antibodies (#1 and #2) expressing the germline VH3-6 allele generated from splenic B-1a cells from 6 wk old mice, described in *Figure 2C*, as measured by flow cytometry. (E) The percentage of heavy chain CDR3 nucleotide sequencing reads expressing the germline VH11-2 allele in peritoneal cavity (PerC) (left) and spleen (right) B-1a samples (% usage). (F) The combined normalized read counts of PtC-binding CDR3 peptide sequences (MRYSNYWYFDV, MRYGSSYWYFDV, and MRYGNYWYFDV) in peritoneal cavity (PerC) (left) and spleen (right) B-1a samples. Black circles represent WT mice, blue circles represent *Tlr2⁻/⁻Tlr4⁻/⁻Unc93b1*^3d/3d mice, green circles represent *Unc93b1*^3d/3d mice, and pink circles represent *Tlr2⁻/⁻Tlr4⁻/⁻* mice (B, C, E, F). CDR3 frequencies were artificially scaled to 10 million reads to account for differences in read depth among samples (F). Error bars indicate the mean (± SEM). *p<0.05, **p<0.01, and ***p<0.001 (One-way ANOVA). Each data point represents an individual mouse (B, C, E, F). Data representative of 3 independent experiments (D). There are three source files associated with this figure.

*Figure 6 continued on next page*

*Figure 6 continued*

DOI: https://doi.org/10.7554/eLife.47015.016

The following source data and figure supplements are available for figure 6:

**Source data 1.** Top 10 highly recurring CDR3 sequences (peptide and V(D)J recombination) detected in peritoneal cavity B-1a samples from age-matched mice lacking different subsets of TLRs.

DOI: https://doi.org/10.7554/eLife.47015.019

**Source data 2.** Top 10 highly recurring CDR3 sequences (peptide and V(D)J recombination) detected in spleen B-1a samples from age-matched mice lacking different subsets of TLRs.

DOI: https://doi.org/10.7554/eLife.47015.020

**Source data 3.** Variable heavy chain gene usage summary in spleen and peritoneal cavity B-1a cells in mice lacking different subsets of TLRs.

DOI: https://doi.org/10.7554/eLife.47015.021

**Figure supplement 1.** VH gene-usage distribution in splenic and peritoneal cavity B-1a samples.

DOI: https://doi.org/10.7554/eLife.47015.017

**Figure supplement 2.** There is considerable variation in B-1a IgH CDR3 repertoire between different mice, independent of TLR signaling.

DOI: https://doi.org/10.7554/eLife.47015.018

generated three monoclonal antibodies from splenic B-1a cells expressing VH3-6, two of which were reactive against the microbiota by mFLOW, further supporting the link between TLR2/4 and microbiota-reactivity (*Figure 6D*). These data suggest that the observed reduction in microbiota-reactive serum IgM in $Tlr2^{-/-}Tlr4^{-/-}$ mice by mFLOW (*Figure 5D*) is at least in part due to the selective reduction of certain specificities from the B-1a repertoire at steady state.

As expected, we observed a significant reduction in VH11-2 gene usage in $Tlr2^{-/-}Tlr4^{-/-}Unc93b1^{3d/3d}$, and $Unc93b1^{3d/3d}$ mice (*Figure 6E*), but not in $Tlr2^{-/-}Tlr4^{-/-}$ mice, consistent with our previous results showing an Unc93B1-dependent loss of PtC-reactive B-1a cells using PtC-liposomes (*Figure 5E–F*). Moreover, the total frequencies of known PtC-reactive CDR3 peptide sequences MRYSNYWYFDV, MRYGSSYWYFD, and MRYGNYWYFDV were also reduced in both peritoneal cavity and splenic B-1a samples from mice lacking Unc93B1, but not in mice lacking TLR2 and TLR4 (*Figure 6F*).

Altogether, these data show that distinct VH genes are differentially regulated by unique subsets of TLRs in B-1a cells—namely, that VH11-2-expressing B-1a cells require Unc93B1-dependent TLRs, whereas VH3-6 and VH9-3 expressing B-1a cells require TLR2/4. Of note, pair-wise comparisons of CDR3 peptide sequences between mice revealed considerable variation in the IgH repertoire within individual mice, independent of genotype (*Figure 6—figure supplement 2*). Therefore, while we were able to observe significant differences in the VH usage patterns in mice lacking subsets of TLRs, it is possible that there are additional TLR-dependent changes to the B-1a repertoire not revealed by these sequencing methods.

## TLR stimulation results in downregulation of CD5 on B-1a cells

Finally, we sought a mechanism for how signaling by TLRs could be linked to activation of B-1a cells with certain specificities. To examine how TLR signaling modulates B-1a activation, we performed an in vitro stimulation of B-1a cells with various TLR ligands. Strikingly, in vitro TLR activation of B-1a cells with TLR2, TLR4, TLR7, and TLR9 agonists resulted in the reduction of surface CD5 (*Figure 7A–B*), a known inhibitor of BCR signaling. These results support a model in which TLR activation renders B-1a cells responsive to BCR stimulation by negatively regulating an inhibitor of BCR signaling. In work co-submitted with this manuscript, Savage and colleagues show that TLR activation of B-1a cells both in vitro and in the context of influenza infection results in the loss of surface CD5 and a concomitant increase in BCR-downstream signaling. Taken together with our previous mFLOW and PtC-liposome results, these data suggest that the specificity of the BCR determines the nature of the TLR ligand encountered by a given B-1a cell. More specifically, we propose a model whereby BCR engagement with the self antigen PtC present on dead or dying cells may facilitate Unc93B1-dependent TLR engagement by 'self' nucleic acids within endosomes, whereas BCR engagement with microbiota-derived antigens may coordinate with PAMP sensing by TLR2 or TLR4 to regulate responses to the microbiota (*Figure 7C*).

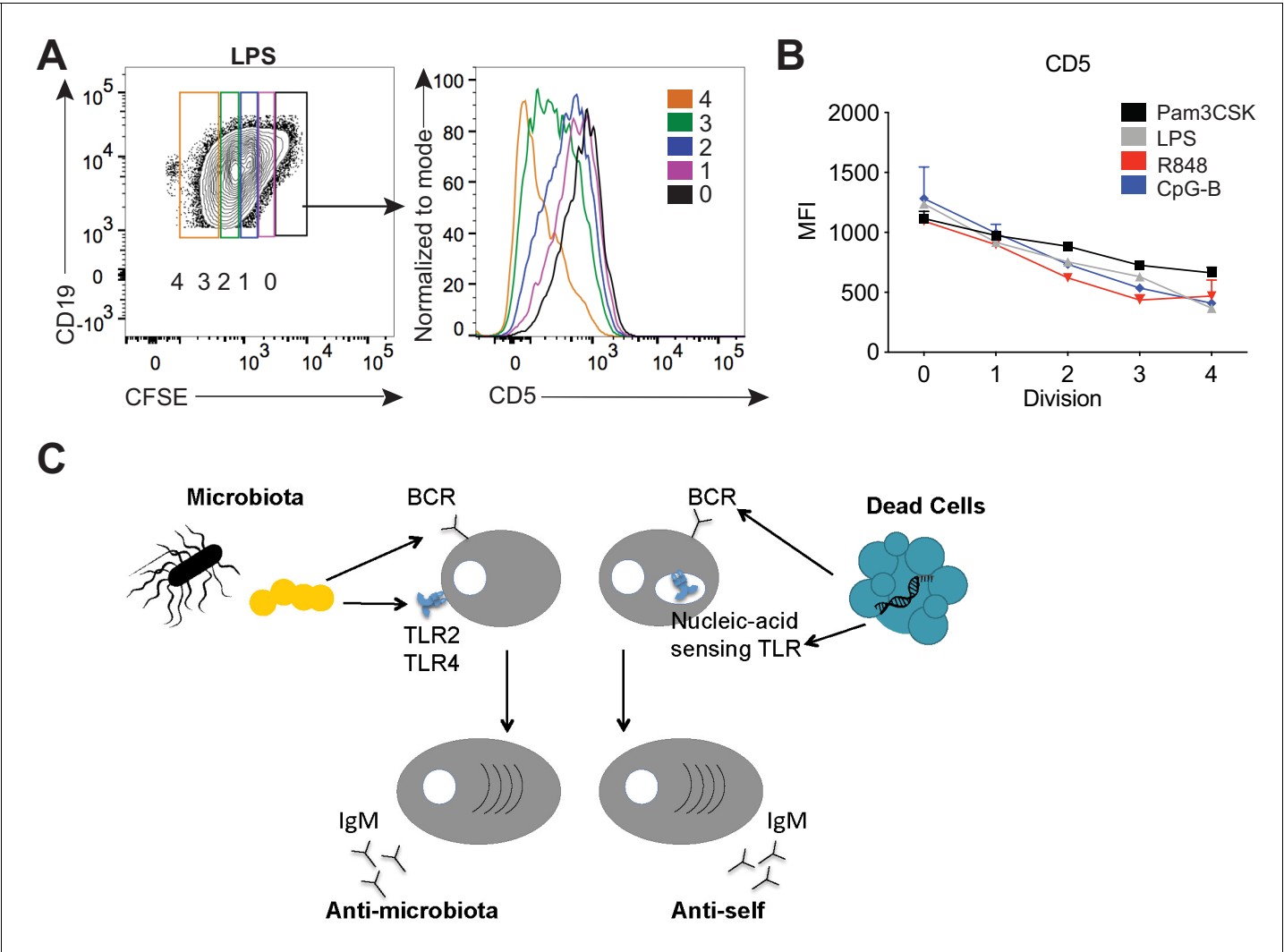

**Figure 7.** TLR stimulation results in CD5 downregulation on B-1a cells. (**A**) Representative flow cytometry histogram plot and quantification of mean fluorescence intensity of CD5 expression (**B**) of in vitro Pam3CSK (black), LPS- (gray), R848- (red) and CpG-B- (blue) stimulated peritoneal cavity B-1a cells at 0, 1, 2, 3, and 4 cycles of division as determined by their CFSE dilution (n = 3, technical replicates). (**C**) Model depicting how TLR and BCR signaling integrate to control distinct B-1a responses. Data are representative of at least three independent experiments (**A-B**).
DOI: https://doi.org/10.7554/eLife.47015.022

## Discussion

The study of B-1a biology has been hampered by a lack of tools to elucidate the signals that regulate B-1a activation and function. In this study, we generated $Ighg3^{T2A\text{-}Cre:TdTomato}$ mice, which serendipitously mark a subset of activated, antibody-secreting IgM+ B-1a cells, allowing us to define the signals that regulate B-1a activation in vivo. The significantly higher expression of the BCR-downstream signaling gene Nur77 in splenic versus peritoneal cavity B-1a cells combined with the enrichment of BCR reactivities to both PtC and the intestinal microbiota in the Tomato+ activated subset of splenic B-1a cells from reporter mice strongly suggests a BCR-mediated expansion process. Interestingly, we did not see Tomato expression in fetal or neonatal liver B-1a cells; rather, Tomato expression in B-1a cells correlated with their entry into the spleen or other distal sites where they likely first encounter antigen. Given the observed reduction in microbiota reactive B-1-derived serum IgM in GF mice, we hypothesize that the microbiota is one such source of antigen. Moreover, we demonstrate that TLR signaling is important for defining the B-1a repertoire and the steady state antibody secretion by B-1a cells, and implicate specific and distinct TLRs for PtC-reactive versus

microbiota-reactive B1a responses. Altogether, our data suggest a model whereby dual BCR and TLR activation regulates the steady state maintenance and antibody secretion of a distinct subset of B-1a cells.

While we found that Tomato expression in *Ighg3*[T2A-Cre:TdTomato] mice correlated with GLT in the *Ighg3* locus, the exact signals that modulate Tomato expression in reporter mice are still not fully understood. Moreover, why there is preferential expression of Tomato in B-1 cells over other B cell subsets is also not known. One likely explanation is that B-1 cells are the primary source of serum IgG3 (*Savage et al., 2017*), and therefore may preferentially receive signals that 'poise' B-1 cells for IgG3 CSR. Additionally, NF-κB binding sites have been identified in enhancers linked to IgG3 CSR (*Michaelson et al., 1996*). Accordingly, *p50*[-/-] activated B cells fail to induce *Ighg3* GLT (*Snapper et al., 1996*). Therefore, BCR and/or TLR signaling, which both result in NF-κB activation, could mediate induction of *Ighg3* GLT (*Snapper et al., 1996*; *Cogné et al., 1994*). The presence of these signals would induce *Ighg3* GLT but not CSR, unless *Aicda* is also expressed. This mechanism is supported by evidence of BCR-mediated activation in Tomato[+] B-1a cells and the ability to induce Tomato expression in *Ighg3*[T2A-Cre:TdTomato] splenocytes activated in vitro with TLR ligands. Future studies involving the re-derivation *Ighg3*[T2A-Cre:TdTomato] mice to GF status would be very interesting, since we would expect the loss of microbiota-reactive Tomato[+] B-1a cells.

Our data also shed light on the role of self versus foreign antigens in shaping B-1a responses. Previous studies have shown that B-1a cell numbers and serum IgM titers are unaltered in GF mice, demonstrating that exogenous antigens are not required for 'natural' serum IgM. IgH sequencing studies in GF and SPF mice have shown that the mature B-1a repertoire is not significantly different in mice lacking a microbiota (*Yang et al., 2015*). We were therefore surprised to observe a significant reduction in microbiota-reactive serum antibodies in GF mice, despite no change in the frequency of PtC reactivity. This finding reveals that a subset of B-1a cells require a microbiota for their expansion and/or antibody secretion.

Like GF mice, we found that mice lacking TLR2 and TLR4 also had a significant reduction in serum IgM reactive against the microbiota. Taken together, these results suggest that the reduction in anti-commensal serum IgM observed in GF mice may be due to the loss of a microbiota-dependent TLR2/4 signal that drives the expansion and antibody secretion of a subset of commensal-reactive B-1a cells. Consistent with previous studies showing no significant differences in the IgH B-1a repertoire in SPF versus GF mice, the VH usage patterns in *Tlr2*[-/-]*Tlr4*[-/-] mice are almost indistinguishable from WT mice. We did, however, find a significant reduction in VH3-6 and VH9-3 gene usage in B-1a cells from mice lacking TLR2 and TLR4. Moreover, monoclonal antibodies generated from WT splenic B-1a cells expressing VH3-6 germline alleles are reactive against the microbiota, suggesting that commensal bacteria may result in the selective expansion and antibody secretion of a subset of B-1a cells through dual BCR and TLR2/4 stimulation. This model is further supported by our results showing the selective reduction in steady state generation of and antibody secretion by PtC-reactive, VH11-2-expressing B-1a cells in mice lacking Unc93B1-dependent TLRs. In this context, BCR engagement with PtC present on dead or dying cells may provide an environment where exposed nucleic acids can be sensed by Unc93B1-dependent TLRs within endosomes. The development of new tools capable of specifically tracking commensal-reactive B cells or the generation of monoclonal antibodies from GF or *Tlr2*[-/-]*Tlr4*[-/-] mice B-1a cells will be important to further elucidate the contribution of a microbiota to the development and/or expansion of commensal-reactive B-1a cells.

Of note, we were surprised that TLR-deficient mice did not have significantly altered serum IgM titers and B-1a frequencies despite a significant reduction in anti-microbiota and PtC reactivity. However, adoptive transfer of TLR-deficient B cells into B-cell deficient mice resulted in ~2–4 fold reduction in total serum IgM titers when compared to mice adoptively transferred with WT B cells, suggesting that a significant source of serum IgM is indeed mediated through TLR activation of B-1 cells. Of note, as evidenced by the incomplete abrogation of serum IgM, microbiota- and PtC- reactive B-1a responses, and otherwise very similar B-1a IgH BCR repertoire in TLR-deficient mice when compared to WT counterparts, other signals, such as cytokines, also likely play a role in regulating TLR-independent B-1a derived antibody responses at steady state. Previous studies have identified IL-5, IL-10, and IFN-γ as important regulators of B-1 responses (*Nisitani et al., 1995*; *Waffarn et al., 2015*). These mechanisms may even compensate for the absence of TLR signaling, resulting in apparently normal total IgM titers and B-1a frequency, despite differences in the B-1a repertoire and specificities of B-1a derived serum IgM.

The presence of a B cell subset that produces self-reactive antibodies contradicts the paradigm that self-reactive clones are deleted in B cell development to avoid autoimmunity. However, B-1a-derived serum IgM likely evolved as an important first line of defense important for both tissue and intestinal homeostasis. Our finding that microbial sensing TLRs regulate anti-microbiota responses, whereas nucleic acid sensing TLRs regulate responses to PtC present on dead or dying cells where DNA or RNA may be sensed, strongly supports a model wherein TLR signaling integrates with BCR signaling to control B-1a responses at steady state. The balance between protective and harmful B-1a responses is likely achieved, in part, through the integration of these two signals. These results could be especially relevant to autoimmune disease where several studies have linked disease severity with hyper-responsive nucleic acid-sensing TLR responses and increased autoantibody production (*Leadbetter et al., 2002*; *Lau et al., 2005*; *Fukui et al., 2009*; *Subramanian et al., 2006*; *Pisitkun et al., 2006*).

Our in vitro results showing that TLR activation of B-1a cells results in downregulation of the BCR inhibitory protein CD5, provide a possible mechanism for how TLRs may render B-1a cells responsive to BCR-mediated activation. Supporting our model of coordinated integration of TLR/BCR signaling in regulating B-1a responses at steady state, Savage et al. (in an accompanying manuscript) found that B-intrinsic TLR signaling is also critical for the differentiation of B-1a cells into antibody secreting cells in the context of influenza infection. Moreover, Savage et al. reveal through biochemical analysis that TLR activation of B-1a cells results in both reduced expression of CD5 and reduced association of CD5 with the IgM-BCR complex, which correlates with an increase in BCR downstream signaling. These results raise important questions regarding the precise molecular mechanism of how TLR and BCR signaling coordinate to control B-1a responses in vivo. Of particular interest is the relative timing of TLR versus BCR signaling on B-1a cells. It is possible that TLR signaling precedes the subsequent ability of B-1a cells to respond to BCR-mediated activation by inducing downregulation of CD5. It is not clear, however, how this mechanism explains our finding that certain B-1a BCR specificities are linked with subsets of TLRs. One possible explanation is that TLR ligands are not present at sufficiently high levels in vivo, so BCR-mediated acquisition of TLR ligands may be required for efficient TLR activation and conversion of B-1a cells into antibody-secreting CD5⁻'ex-B-1a' B-1b cells. In vitro stimulation with high concentrations of TLR ligands may bypass this requirement. Future studies using conjugates of antigens and TLR ligands could enable a more precise delineation of the sequence of events leading to B-1a proliferation and antibody secretion.

## Materials and methods

### Data availability

All data generated or analyzed during this study are included in the manuscript and supporting files. Source data files have been provided for *Figures 2* and *6*. Data files associated with RNA sequencing experiments of B-1a cells have been deposited on Dryad (DOI: 10.5061/dryad.660js04).

### Mice

C57BL/6, μMT$^{-/-}$ (002288), *Nur77*$^{GFP}$ (016617), *Rosa26*$^{STOP-flox-TdTomato}$ (007905), *Rosa26*$^{STOP-flox-DTA}$ (009669), β-actin$^{Fippase}$ (005703) were obtained from The Jackson Laboratories. *Ighg3*$^{T2A-Cre}$ mice were generated by constructing a targeting vector encoding the self-cleaving peptide T2A followed by the *Cre recombinase* gene to be inserted immediately 3' of the last *Ighg3* transmembrane exon. This vector was electroporated into C57BL/6-derived embryonic stem cells by the Mouse Biology Program at UC Davis. The vector also introduced a *frt*-flanked *neomycin resistance* gene. Targeting was assessed by Southern blot and correctly targeted ES cells were injected into ICR/CD1 blastocysts. Chimeric males were mated with C57BL/6 background β-actin$^{Fippase}$ females to remove the neomycin resistance cassette. Mice were then crossed to either *Rosa26*$^{STOP-flox-TdTomato}$ to permanently mark Cre$^+$ cells with TdTomato or to *Rosa26*$^{STOP-flox-DTA}$ to ablate Cre$^+$ cells. *Tlr2*$^{-/-}$ and *Tlr4*$^{-/-}$ mice were generated and provided by S. Akira (Osaka University) (*Takeuchi et al., 1999*; *Hoshino et al., 1999*). *Unc93b1*$^{3d/3d}$ were purchased from MMRCC (*Tabeta et al., 2006*). All mice were bred and maintained in specific pathogen-free conditions at UC Berkeley. Germ-free C57BL/6 mice were maintained at the UC Berkeley Germ-Free Facility. Experiments were performed in strict accordance with the recommendations in the Guide for the Care and Use of Laboratory Animals of

the National Institutes of Health. All of the animals were handled according to approved institutional animal care and use committee (IACUC) protocol (#2017-03-9679) of the University of California, Berkeley and IACUC protocol (#18797) of the University of California, Davis.

For co-housing experiments, at the time of weaning (3 weeks of age) mice were co-housed for at least 4 weeks prior to sacrifice to normalize their microbiota.

## Cell lines

HEK293T cells were obtained from American Type Culture Collection (ATCC CRL-11268, Manassas, VA). Authentication of the cell line was performed by STR profiling and had a negative mycoplasma contamination testing status. Cells used in experiments were below 10 passages.

## Adoptive transfer experiments

To generate neonatal B-1 cell chimeras, newborn B6.Cg IgH$^a$ Thy1$^a$ Gpi1$^a$ /J (IgH$^a$) (Jackson, 001317) were treated with 2 mg per mouse of anti-IgM$^a$ (clone DS1.1) intraperitoneally twice weekly for 6 weeks to temporarily deplete B cells. 2–3 days after birth, pups were reconstituted with peritoneal lavage cells from C57BL/6 (IgH$^b$), as a source of wild-type allotypically marked IgH$^b$ B-1 cells. Mice were analyzed at least 2 months after the end of anti-IgM$^a$ treatment, where B-2 cells were host derived (IgH$^a$) and the majority of B-1 cells were derived from the donor peritoneal cavity (IgH$^b$).

For adoptive transfer of either WT or $Tlr2^{-/-}Tlr4^{-/-}Unc93b1^{3d/3d}$ B cells, CD19$^+$ cells from the bone marrow and spleen were isolated by using biotinylated antibodies and Streptavidin conjugated to magnetic beads (MACS Miltenyi), and $5 \times 10^6$ cells were retro-orbitally transferred into 2 day old µMT$^{-/-}$ neonates. Mice were analyzed 10 weeks after transfer.

## Cell isolation and in vitro stimulation assays

Peritoneal cavity cells were isolated by peritoneal lavage with 5 mL PBS. Bone marrow cells were isolated from a single femur and red blood cells were lysed using Ammonium-Chloride-Potassium (ACK) buffer (Gibco). Spleens were digested with collagenase 8 (Sigma) and DNAse-I for 45 min and red blood cells were lysed using Ammonium-Chloride-Potassium (ACK) buffer (Gibco). For stimulation assays, 200,000 cells were incubated in media (RPMI/10%FCS/L-glutamine/Pen-Strep/HEPES/Sodium pyruvate/βME) with LPS (25 µg/mL) in 96-well round-bottom plates for 72 hr. Flow cytometry was used to analyze stimulated cells.

## Southern blotting

Murine embryonic stem cell DNA was screened for single-copy targeted insertion of *T2ACre* in the *Ighg3* locus by digesting DNA with BglII restriction endonuclease, transferring agarose-gel run digested DNA onto a nitrocellulose membrane, and hybridizing membrane with P-32 labeled DNA probe specific for the 5' UTR of *Ighg3* (forward primer: 5'-TGAGCCAGGGTAAGTGGGAGTATG-3'; reverse primer: 5'-ATGAGGTGCAGAGTGGCTACAGG-3') and the Neomycin resistance gene (forward primer: 5'-ACCGTAAAGCACGAGGAAGCG-3'; Reverse primer: 5'-GCACGCAGGTTC TCCGGC-3'). Southern blots were visualized with the use of a Typhoon scanner (Amersham Biosciences).

## Flow Cytometry

Single-cell suspensions from the spleen and peritoneal cavity were stained with anti-CD16/32 for 20 min at 4 °C to block Fc receptors. Cells were stained with the following conjugated antibodies: anti-CD19 (eBio; clone: 1D3; BV711 or PECy7), anti-IgG3 (BD; clone R40-82; FITC), IgM (eBio; clone: 11/41; APC-780), anti-CD5 (eBio; clone: 53–7.3; APC), anti-CD43 (BD; clone: S7; PE or BV421), anti-IgD (eBio; clone: 11–26 c; PECy7), anti-CD23 (eBio; clone: B3B4; PerCP-710), anti-CD21/35 (eBio; clone: eBio8D9; APC-780), anti-IgM$^b$ (BD; clone: AF6-78; BV650), anti-Nur77 (BD; clone: 12.14; PerCP-710), anti-CD4 (BD; clone: L3T4; Biotin or FITC), anti-CD3e (eBio; clone: 145–2 C11; Biotin or FITC), anti-CD8 (Tonbo; clone: 53–6.7; Biotin or FITC), anti-NK1.1 (eBio; clone PK136; Biotin or FITC), anti-Ly6-G (eBio; clone: RB6-8C5; Biotin or FITC), anti-F4/80 (eBio; clone: BM8; Biotin or FITC), or streptavidin (eBio; conjugated to FITC). Dead cells were excluded by DAPI viability dye and near-IR fixable viability stain for experiments where cells were fixed and permeabilized. Flow cytometry was

performed using an LSRFortessa (BD Biosciences) and cell sorting was performed using a FACSAria (BD Biosciences).

For intranuclear staining for Nur77, cells were surface stained for 30 min at 4 °C prior to fixation (Thermo Fisher) for 1 hr at room temperature. Cells were washed and stained with anti-Nur77 for 1 hr at room temperature.

For fluorescein-labeled PtC liposome flow cytometry, cells were co-incubated with PtC-Liposomes and surface stain for 1 hr at 4 °C. PtC-liposomes were prepared by modification of a previously described method (*Mercolino et al., 1988*). Briefly, liposomes were prepared by high pressure extrusion though a final filter pore size of 0.05 microns and consisted of DSPC:DSPG:Chol in a molar ratio of 45:5:50 encapsulated with fluorescein sulfonic acid (5 mM). For some experiments Oregon Green 488-DHPE Liposomes (DOPC/CHOL/OG-PE (54:45:1) were used (FormuMax, Sunnyvale, CA).

## Microbiota flow cytometry (mFLOW)

For microbiota flow cytometry, a fecal pellet from a B-cell deficient µMT$^{-/-}$ mouse from our mouse colony was resuspended in sterile-filtered PBS and spun at 200 rcf for 5 min. Supernatant was transferred to a new sterile tube and washed 2 times with PBS by spinning at 8,000 RPM for 5 min. OD$_{600}$ of suspension was measured, and bacteria was diluted in 1% bovine serum albumin (BSA; Fisher) at final OD$_{600}$ of 0.01. Mouse serum was diluted 1:25 in PBS/BSA buffer, and 25 µl of this solution was added to 25 µl diluted fecal bacteria in a 96-well v-bottom plate, and incubated overnight at 4C. For hIgG1 microbiota flow, a fecal pellet from the same mouse from which recombinant hIgG1 monoclonal antibodies (mAbs) were generated was stained with 4 µg/mL hIgG1 mAbs overnight. mAbs that did not yield above 4 ug/mL antibody by ELISA were excluded from microbiota flow analysis. Staining was performed with fluorochrome conjugated anti-IgM (eBio; clone: 11/41; Biotin), anti-IgM$^b$ (BD; clone: AF6-78; BV650), anti-IgM$^a$ (BD; clone: DS-1; BV711), or goat anti-hIgG1-biotin (Jackon Immunoresearch) followed by streptavidin-PECy7 (eBio) when using biotinylated primary antibodies. Cells were washed and resuspended in SYBR Green (Invitrogen) and analyzed by FACS using an LSRFortessa (BD Biosciences). For analyses, Sybr$^+$ events were defined as the population of Sybr$^{hi}$ events not present in GF mouse feces.

## Recombinant hIgG1 monoclonal antibody generation

hIgG1 monoclonal antibodies were generated from single-cell sorted Tomato$^-$ or Tomato$^+$ IgM$^+$ splenic B-1a cells (CD19$^+$CD23$^-$CD5$^+$CD43$^+$IgM$^+$) from a 6 week old *Ighg3*$^{T2A-Cre:TdTomato}$ mouse according to previously described protocol (*Tiller et al., 2009*). hIgG1 and hkappa expression plasmids have been previously described (*Smith et al., 2009*). HEK293T cells were obtained from American Type Culture Collection (ATCC, Manassas, VA). Productively rearranged paired heavy and light chain sequences were included in analysis of VH gene usage comparisons between Tomato$^-$ and Tomato$^+$ mAbs.

## Single-cell RT-PCR of *Ighg3* germ-line transcript (GLT)

Tomato$^+$IgM$^+$IgG3$^-$CD19$^+$ cells or Tomato$^+$IgM$^-$IgG3$^+$CD19$^+$ from *Ighg3*$^{T2A-Cre:TdTomato}$ LPS stimulated splenocytes were single cell sorted into 96-well plates and cDNA was synthesized as previously described (*Tiller et al., 2009*), with the only deviation being the use of iScript (Thermo Scientific). Single-cell semi-nested RT-PCR for *Ighm* mRNA, *Ighg3* mRNA, and *Ighg3* GLT was performed and products were run on agarose gel for visualization.

## ELISA and ELISPOT Assays

Nunc Hi Affinity ELISA plates were coated with goat anti-murine IgM (500 ng/mL; Jackson Immunoresearch) or human kappa chain (1 µg/mL; MyBioSource), and blocked with PBS with 1% BSA (w/v) and 2% goat serum (Gibco; v/v). Secondary peroxidase conjugated goat antibodies to murine IgM or to human IgG (Jackson Immunoresearch) were used at 1:5000 in PBS. Purified murine IgM and human IgG1 standards were from eBioscience and Sigma, respectively. Plates were developed with 1 mg/mL OPD (Sigma) in Citrate Buffer (PBS with 0.05M NaH2PO4 and 0.02M Citric acid) with 3M HCl acid stop. Absorbance at 490 nm was measured on a SpectraMax M2.

For ELISpot analysis, multiscreen plates (Millipore) were coated with goat isotype-specific antibodies to murine IgM (5 µg/mL; Jackson Immunoresearch) in PBS and blocked with PBS with 1%

BSA (w/v) and 5% goat serum (v/v). Plates were washed with PBS and cells were serially diluted in complete RPMI and incubated at 37°C overnight. Following several washes in PBS, secondary peroxidase conjugated antibodies to IgM (Jackson Immunoresearch) were used at 1:1000 in PBS to detect antibody-secreting cells. Plates were developed with AEC developing reagent (Vector Laboratories) according to manufacturer's instructions. Plates were read on an ImmunoSpot C.T.L. Analyzer (v3.2) and quantitated using ImmunoSpot 5.1 Pro.

### Immunoglobulin heavy chain CDR3 RNA sequencing and analysis

Splenocytes were depleted of $CD3^+CD4^+CD8^+NK1.1^+F4/80^+GR-1^+$ cells using biotinylated antibodies and Streptavidin conjugated to magnetic bead MACS Miltenyi magnetic bead depletion prior to sorting. $CD19^+IgM^{hi}IgD^{lo}CD43^+CD5^+$ depleted spleen and peritoneal cavity lavage cells were sorted using a FACS Aria (BD Biosciences). Sorted cells were immediately resuspended in TRIzol (Invitrogen) and snap frozen. Samples were subsequently processed using the services provided by iRepertoire Inc. Briefly, bulk RNA was extracted, heavy chain cDNA libraries were generated, and CDR3 nucleotide sequences were amplified and sequenced using paired-end MiSeq sequencing technology. Regular un-normalized or normalized CDR3 nucleotide sequencing data were analyzed. For CDR3 length and N-nucleotide addition analysis, each unique CDR3-VDJ combination was treated as a quantity of one regardless of read count, allowing us to view the repertoire independent of skewing which may occur to just one or a few highly dominant clones. Because our analysis was performed on sorted B-1a cells, excluding plasma cells, which do not express CD5, the frequency of a given Ig transcript is roughly viewed as the relative index of frequency of cells expressing this Ig transcript. Additionally, clonal expansion starting at around the time of weaning is a hallmark of self-replenishing B-1a cells. Therefore, for VH gene usage frequencies, we did not normalize the data by scoring each distinct IgH CDR3 nucleotide sequence expressing a given VH gene as one, independent of how often this sequence was detected, as is often done with bulk RNA-based Ig sequencing methods. For VH gene usage frequencies and pair-wise comparisons of percentage shared CDR3 sequences between samples, total read counts were normalized to 10 million reads to account for differences in read depth between samples. A minimum frequency cutoff of 1000 was applied to pair-wise comparison analysis.

### Statistical analysis

All data are presented as mean (± SEM). Statistical analysis was done using paired or unpaired two-tailed Student's t-test for comparisons between two groups, or one-way ANOVA for multiple comparisons. $p < 0.05$ was considered significantly different.

## Acknowledgements

We thank members of the Barton and Vance Labs for helpful discussions and critical reading of the manuscript. We thank Hector Nolla and Alma Valeros for assistance with cell sorting at the Flow Cytometry Facility of the Cancer Research Laboratory at UC Berkeley and Dr. David Rawlings for sharing hIgG1 and hkappa recombinant antibody expression plasmids. This work was supported by the NIH (AI072429 and AI063302 to GMB; T32 GM007232 and T32 AI100829 to LSMK) and by an Investigator in the Pathogenesis of Infectious Disease award to GMB from the Burroughs Wellcome Fund.

## Additional information

### Funding

| Funder | Grant reference number | Author |
|---|---|---|
| National Institute of Allergy and Infectious Diseases | AI142926 | Lieselotte SM Kreuk<br>Meghan A Koch<br>Leianna C Slayden<br>Nicholas A Lind<br>Sophia Chu<br>Gregory M Barton |

| National Institute of Allergy and Infectious Diseases | AI063302 | Lieselotte SM Kreuk<br>Meghan A Koch<br>Leianna C Slayden<br>Nicholas A Lind<br>Sophia Chu<br>Gregory M Barton |
|---|---|---|
| National Institute of General Medical Sciences | GM007232-36 | Lieselotte SM Kreuk |
| National Institute of Allergy and Infectious Diseases | AI100829 | Lieselotte SM Kreuk |

The funders had no role in study design, data collection and interpretation, or the decision to submit the work for publication.

## Author contributions

Lieselotte SM Kreuk, Conceptualization, Data curation, Formal analysis, Supervision, Validation, Investigation, Visualization, Methodology, Writing—original draft, Project administration, Writing—review and editing; Meghan A Koch, Conceptualization, Methodology, Writing—review and editing; Leianna C Slayden, Investigation, Methodology, Writing—review and editing; Nicholas A Lind, Investigation, Writing—review and editing, Generated monoclonal antibodies and performed key experiments for manuscript; Sophia Chu, Investigation, Performed key in vivo experiments for manuscript; Hannah P Savage, Resources, Investigation, Methodology, Performed key B-1 chimera experiments; Aaron B Kantor, Resources, Methodology, Writing—review and editing, Provided PtC-liposome reagent and methodology for use in flow cytometry; Nicole Baumgarth, Conceptualization, Resources, Investigation, Writing—review and editing; Gregory M Barton, Conceptualization, Resources, Data curation, Supervision, Funding acquisition, Methodology, Writing—review and editing

## Author ORCIDs

Lieselotte SM Kreuk https://orcid.org/0000-0003-4906-039X
Nicole Baumgarth http://orcid.org/0000-0002-2891-4483
Gregory M Barton https://orcid.org/0000-0002-3793-0100

## Ethics

Animal experimentation: Experiments were performed in strict accordance with the recommendations in the Guide for the Care and Use of Laboratory Animals of the National Institutes of Health. All of the animals were handled according to approved institutional animal care and use committee (IACUC) protocol (#2017-03-9679) of the University of California, Berkeley and IACUC protocol (#18797) of the University of California, Davis.

## Decision letter and Author response

Decision letter https://doi.org/10.7554/eLife.47015.027
Author response https://doi.org/10.7554/eLife.47015.028

# Additional files

## Supplementary files

• Transparent reporting form
DOI: https://doi.org/10.7554/eLife.47015.023

## Data availability

All data generated or analyzed during this study are included in the manuscript and supporting files. Source data files have been provided for Figures 2 and 6. Data files associated with RNA sequencing experiments of B-1a cells are available from the Dryad Digital Repository: https://dx.doi.org/10.5061/dryad.660js04.

The following dataset was generated:

| Author(s) | Year | Dataset title | Dataset URL | Database and Identifier |
|---|---|---|---|---|
| Kreuk LSM, Koch MA, Slayden LC, Lind NA, Chu S, Savage HP, Kantor AB, Baumgarth N, Barton GM | 2019 | Data from: B cell receptor and TLR signaling coordinate to control distinct B-1 responses to both self and the microbiota | https://dx.doi.org/10.5061/dryad.660js04 | Dryad Digital Repository, 10.5061/dryad.660js04 |

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
