## [Decision Letter]

Thank you for sending your article entitled "B cell receptor and TLR signaling coordinate to control distinct B-1 responses to both self and the microbiota" for peer review at *eLife*. Your article is being evaluated by three peer reviewers, and the evaluation is being overseen by a Reviewing Editor and Wendy Garrett as the Senior Editor.

Thank you for your patience during the evaluation process. Since your paper was submitted in parallel with the manuscript "TLR-signaling induces reorganization of the IgM-BCR complex regulating B-1 cell responses to infections" from Baumgarth and colleagues, we aimed to secure two reviews of both manuscripts (Nos. 1 and 3) and to carry out the consultation in parallel.

We believe that the work addresses an extremely important problem of the coordination of B cell receptor and TLR signaling, although a major concern of reviewer 2 (who had not seen the other manuscript) was the evidence in this paper for the B cell intrinsic TLR effects. The 2 rounds of consultation came to the view that although there is no exact complementarity between the two papers in terms of antigens examined and experimental systems, a combination of cross-referencing between the papers and express caveats for the evidence on this point in your manuscript may address the issue without necessarily requiring in vivo chimeras. We do not think that further germ-free colonization experiments are in the current scope.

*Reviewer #1:*

In this manuscript, Barton et al. describe the serendipitous development of a B1 B cell reporter mouse and the subsequent use of this mouse (and derivations of) to interrogate the mechanisms that lead to the development of self-reactive and commensal-reactive IgM. Most notably, they identify unique contributions of nucleic acid-sensing versus bacterial-sensing TLRs to the emergence of these two antibody specificities, respectively. Overall, this is an elegant and nicely-done study that addresses long-standing issues regarding B1 cell biology and reveals a new mechanism of integration of innate and adaptive signals in B cells themselves. I only have one major experimental concern, which revolves around the data arguing for a role of TLR2/4 in commensal reactivity. It seems highly likely that gut microbiota composition (particularly the presence or absence of pathobionts like Helicobacter or others) could influence the development of commensal-specific IgM. Thus, different mouse colonies (or even lines within a facility that are bred separately) might exhibit differing levels of anti-commensal IgM based merely on differences in commensal composition.

Unless I am mistaken, it appears that the authors performed nearly all of their experiments using mice that were bred as lines rather than using littermate controls, which would be the proper standard in this case for determining the effect of a gene(s) on the observed phenotype. I am certainly sympathetic to the fact that breeding lines with multiple alleles creates a challenge here.

Nonetheless, I would have hoped that the authors would have at least co-housed the different lines to partially correct for potential effects of the microbiota, but based on the methods it seems that these lines were all bred and housed separately. There is even an outside possibility that microbiota-dependent differences could explain the seeming discrepancy between the authors' data and the recently published data by Yang showing that GF and SPF mice have equivalent B1a repertoires-that is, that the Yang experiments were done with a microbiota that failed to induce the specificities that Barton et al. discover here. I would strongly suggest that the authors perform at least a few of the critical experiments (e.g., Figure 5D and E) using littermates or (at minimum) comparing mice that are co-housed to normalized the microbiota. Even more interesting, but likely beyond the scope of this manuscript, would be to colonize GF mice with different microbes or communities and then evaluate the effects on commensal-specific IgM.

*Reviewer #2:*

The crosstalk of TLRs and the BCR on B1 cells is still poorly understood. Kreuk et al. generated a novel mouse strain by targeting Cre into the constant region of Igg3. In combination with a Rosa26-Tomato allele, this strain was used as a reporter mouse. Tomato was found to be expressed in IgG3+ and (predominantly) IgG3- B cells, with the highest frequencies in peritoneal B-1a cells. Splenic Tomato+ B-1a cells showed an increased cell size and enhanced spontaneous antibody secretion compared to their Tomato- counterpart, along with increased frequencies of PtC-specific cells. Moreover, the percentage of splenic Tomato+ B-1a increased with age, which the authors connect to the acquisition of microbiota. Based on transfer and depletion experiments, the authors conclude that the microbiota-reactive IgMs are B1-derived and that their presence in the serum is microbiota-dependent. In a second set of experiments, the authors show that a smaller fraction of the microbiota is bound by IgMs derived from TLR-deficient mice compared to wild-type controls, and that the fraction of splenic and peritoneal PtC-specific B-1a cells is reduced in these animals. By BCR repertoire sequencing, the authors show that the usage of some VH segments is altered in the absence of certain TLR subsets. Overall, the study contains interesting results, but needs to be improved (and restructured) significantly, as pointed out below.

- As the authors are well aware, TLRs are expressed on many cell types. Thus, the phenotype observed in TLR germline-knockout animals may be due to TLR-deficiency on other cell types, i.e. not B-cell intrinsic. As the paper addresses B cell-intrinsic BCR-TLR cross-talk, it is essential that the authors clarify this issue by e.g. mixed bone marrow reconstitution experiments.

- The data on the B1 cell transfer experiments do not include a phenotypic analysis of the B cell subsets in the recipients after depletion of endogenous B cells and at the time of serum isolation. Did the B-cell depletion work? Do the donor cells solely give rise to B1a cells in the recipients? These controls are essential and should be included.

- The Igg3-Cre mouse line is presented as a reporter line for activated B-1a cells. However, a complete characterization of the mouse (B cell development, expression of Tomato in various B cell subsets (including pro-B, pre-B, MZ, B1-b, etc.), comparison of B cell subset frequencies and absolute cell numbers to wild-type counterparts, etc.) is missing and should be provided, as this is a new, previously unpublished transgenic mouse line. Furthermore, how B cell development was affected when Igg3-Cre expressing cells were eliminated through DTA might be very informative and thus worth to be included in the manuscript.

- A substantial fraction of Tomato+ B cells is negative for IgM and IgG3. Do these cells represent plasma cells, or what else do they represent?

- The subset of Tomato+ B-1a cells should be characterized in more detail (e.g. by analyzing B1-typical markers, responses to stimuli, etc.). Furthermore, given that B-1a cells predominantly reside in the peritoneal cavity, the analysis of Tomato+ cells should not be restricted to the spleen, as it appears to be the case in some figures (such as Figure 2H).

- It is an obvious deficit of the paper that Tomato+ cells are not analyzed under GF conditions. This may be technically impossible to achieve, but the authors may comment in the text.

- Some conclusions of the authors are overstatements, such as the title of Figure 4 ("Toll-like receptor signaling is required for B-1a responses to both phosphatidylcholine and the microbiota"). In reality, microbiota reactive IgM and PtC-specific B-1a cells are reduced, but not absent in the TLR knockout animals. Thus, TLRs may be involved but not required. The same applies for the title of Figure 6: Most CDR frequencies in B-1a cells are not affected by the absence of TLRs in the mutant mice.

*Reviewer #3:*

The authors generated IgG3-Tomato reporter mouse and observed that Tomato expression was preferentially found in IgM-secreting, activated fraction of B1a cells. The Tomato+ B1a cells express high levels of Nur77, and the IgM generated by Tomato+ cells efficiently recognize PtC and fecal microbiota. These results show that Tomato+ B1a cells received BCR-signal upon their differentiation, probably through PtC and microbiota. Microbiota binding serum IgM was produced by B1, but not by B2 cells, and the differentiation of this fraction of B1a cells required microbiota colonization and also TLR2/4 signals. Interestingly, the differentiation of PtC reactive B1a cells require the function of Unc93B1 but not TLR2/4 signals. Thus, the authors proposed the model in which the responses of B1a cells to microbiota and self-antigen PtC are regulated by coordinated signals through the BCR and the distinct TLR signaling. This is very interesting study based on their serendipitous observation with IgG3-Tomato reporter mouse. The experiments are well-done and the manuscript is well-written.

Some confusion come from the data in which normal amount of IgM was detected in the serum of GF and TLR-/- animals (Figure 3DFigure 4CFigure 5C). Since Tomato+B1a cells are not present in day0 neonatal liver and accumulate upon microbiota colonization (Figure 2F-H), germ free mice are expected to have fewer amount of Tomato+B1a cells. If so, IgM-secreting B1a cells should be fewer in GF and TLR-/- animals. This can be tested by ELISPOT assay for sorted B1a cells in PerC and SPL of GF and TLR-/- mice, like the authors show in Figure 1G. The normal amount of serum IgM gives the impression that there is no defect on Tomato+B1a cells in GF and TLR animals, which gives confusion and disturbance for interpretation of the results related with these animal models.

Another confusion comes from the data by showing the in vitro down-regulation of CD5 upon TLR stimulations (Figure 7), because the authors suggest that TLR stimulation "induce" the IgM-secreting B1a cells. Especially, Figure 4E directly shows that CD5^+^ B1a cells were reduced in TLR-/- mice. This is confusing, because if TLR stimulation down-regulate CD5 expression, the numbers of CD5^+^ B1a cells should be increased in TLR-/- mice. One possible explanation is that the strength of TLR signal (or amount of the TLR ligands) might be totally different for in vivo B1a induction and for in vitro CD5 down-regulation. The authors can test this possibility by injecting TLR ligands into WT and TLR-/- mice, and compare the numbers of CD5^+^B1a cells, like in Figure 4E.

---

## [Author Response]

Reviewer #1:

[…] *I only have one major experimental concern, which revolves around the data arguing for a role of TLR2/4 in commensal reactivity. It seems highly likely that gut microbiota composition (particularly the presence or absence of pathobionts like Helicobacter or others) could influence the development of commensal-specific IgM. Thus, different mouse colonies (or even lines within a facility that are bred separately) might exhibit differing levels of anti-commensal IgM based merely on differences in commensal composition.*

Unless I am mistaken, it appears that the authors performed nearly all of their experiments using mice that were bred as lines rather than using littermate controls, which would be the proper standard in this case for determining the effect of a gene(s) on the observed phenotype. I am certainly sympathetic to the fact that breeding lines with multiple alleles creates a challenge here. Nonetheless, I would have hoped that the authors would have at least co-housed the different lines to partially correct for potential effects of the microbiota, but based on the methods it seems that these lines were all bred and housed separately. There is even an outside possibility that microbiota-dependent differences could explain the seeming discrepancy between the authors' data and the recently published data by Yang showing that GF and SPF mice have equivalent B1a repertoires-that is, that the Yang experiments were done with a microbiota that failed to induce the specificities that Barton et al. discover here. I would strongly suggest that the authors perform at least a few of the critical experiments (e.g., Figure 5D and E) using littermates or (at minimum) comparing mice that are co-housed to normalized the microbiota. Even more interesting, but likely beyond the scope of this manuscript, would be to colonize GF mice with different microbes or communities and then evaluate the effects on commensal-specific IgM.

We agree with the reviewer that this is a very important control, since it is formally possible that lacking TLRs drives changes in the microbiota, resulting in altered B-1a responses. Therefore, in order to normalize the microbiota between different strains of mice, we cohoused WT and TLR KO mice for 4 weeks after weaning, prior to analyzing PtC and microbiota-reactive B-1a responses. We observed the same reduction in anti-PtC and anti-microbiota B-1a responses in TLR KO mice when cohoused with WT mice as we show in Figure 4, allowing us to conclude that the observed differences we see in B-1a responses in TLR KO mice are not due to alterations in their microbiota but rather due to the loss of TLR signaling. We have included these data as a supplement for Figure 4 (Figure 4—figure supplement 2).

Reviewer #2:

[…] Overall, the study contains interesting results, but needs to be improved (and restructured) significantly, as pointed out below.- As the authors are well aware, TLRs are expressed on many cell types. Thus, the phenotype observed in TLR germline-knockout animals may be due to TLR-deficiency on other cell types, i.e. not B-cell intrinsic. As the paper addresses B cell-intrinsic BCR-TLR cross-talk, it is essential that the authors clarify this issue by e.g. mixed bone marrow reconstitution experiments.

We agree with reviewer 2 that it is important to address the B-cell intrinsic nature of TLR-dependent B-1a responses. Additionally, we appreciate reviewer 3’s comment concerning the confusion presented by the unaltered serum IgM titers in TLR KO mice, given that the data presented in the manuscript supports a model whereby TLR KO B-1a cells secrete less IgM. In order to address these two points in parallel, we performed WT versus TLR KO B cell transfers into B-cell deficient neonates and measured serum IgM responses 10 weeks post transfer. While we had originally proposed including analysis of WT vs. TLR KO B-1 chimeras (generated in collaboration with Baumgarth’s group) to address the question of whether the requirement for TLR function is B cell intrinsic, we were unable to use these data due to technical issues related to the chimera generation.. Fortunately, in parallel we had performed B cell transfers (WT or TLR KO) into B cell deficient mice, as an independent approach to address this question. These experiments clearly demonstrate that B-cell intrinsic TLR signaling is required for optimal antibody secretion and anti-microbiota reactivity (Figure 5K, L). We have included the results from the transfer experiments in our revised manuscript and hope that the Editor and reviewers agree that these data are sufficient to address the B cell intrinsic issue.

We observed significantly reduced microbiota-reactive serum IgM in mice that received TLR KO B cells when compared to WT B cell transfer recipients, allowing us to conclude that TLR-dependent microbiota-reactive B-1 responses are B-cell intrinsic. Additionally, we also observed significantly lower total serum IgM titers in TLR KO B cell transfer recipients. Importantly, while it is not statistically significant, we do see a trend of lower total serum IgM in global TLR deficient animals (Figure 4C). We therefore speculate that the loss of TLRs on other cell subsets may contribute to elevated IgM levels, masking the requirement of B-cell intrinsic TLR signaling for IgM secretion revealed by these B-cell transfer experiments. We have added discussion of this point of confusion in the main text, and have added WT vs. TLR KO B cell transfer data to Figure 4 (Figure 4H-L).

- The data on the B1 cell transfer experiments do not include a phenotypic analysis of the B cell subsets in the recipients after depletion of endogenous B cells and at the time of serum isolation. Did the B-cell depletion work? Do the donor cells solely give rise to B1a cells in the recipients? These controls are essential and should be included.

While we do not have cellular data for the B-1 chimeras upon sacrifice, we have added ELISA data that show endogenous B-1 cell depletion efficiency. Specifically, we have added a panel to Figure 3 (Figure 3B) showing serum IgM^a^ (endogenous B-2 derived) versus IgM^b^ (Donor B-1 derived) titers in the B-1 chimeras presented in Figure 3, which shows that essentially all of the serum IgM is B-1 derived (consistent with previously published data on B-1 Chimeras). This readout is a proxy for endogenous B-1 cell depletion efficiency, as IgM^a^ titers negatively correlate with endogenous B-1 depletion efficiency. We have added to the main text of the manuscript a more detailed description of the B-1 chimera system, which is an extensively validated approach originally published in detail by Lalor et al., 1989. Cellular analyses of B-1 chimeras are further described by Baumgarth et al., 1999. In short, transfer of either purified B-1 cells or total PerC lavage results in only B-1 cells surviving for longer than a few days (i.e. the transferred B-2 cell do not survive post transfer). We hope that the combination of serum ELISA data, which we have added to Figure 3, and referencing previous studies validating the B-1 chimera model in the main text of the manuscript will be sufficient to address this point.

As pointed out by reviewer 2, a PerC lavage contains both CD5^+^ B-1a (~80%) and CD5^–^ B-1b (~20%) cells, which are both self-renewing. However, as shown in the accompanying manuscript from Savage et al., CD5^+^ B-1a differentiate into IgM-secreting CD5^–^ “ex B-1a” B-1b cells, whereas CD5^–^ PerC B-1b cells do not. Moreover, while in the context of influenza infection, Savage et al. generated B-1 chimeras with either purified CD5^+^ B-1a cells or purified CD5^–^ B-1b cells and show that B-1a cells are the main source of IgM ASCs (whereas transferred B-1b cells were not). We therefore feel it is appropriate to conclude that serum IgM derived from transferred PerC lavage cells are largely of B-1a origin.

- The Igg3-Cre mouse line is presented as a reporter line for activated B-1a cells. However, a complete characterization of the mouse (B cell development, expression of Tomato in various B cell subsets (including pro-B, pre-B, MZ, B1-b, etc.), comparison of B cell subset frequencies and absolute cell numbers to wild-type counterparts, etc.) is missing and should be provided, as this is a new, previously unpublished transgenic mouse line. Furthermore, how B cell development was affected when Igg3-Cre expressing cells were eliminated through DTA might be very informative and thus worth to be included in the manuscript.

We have included a number of new figure panels to address the reviewer’s request for a more complete characterization of the *Igg3*^T2A-Cre^ knock-in mouse. We have included two additional supplementary figures to Figure 1 (Figure 1—figure supplements 4 and 5) where we compare C57BL/6, *Igg3*^T2A-Cre:TdTomato^, and *Igg3*^T2A-Cre:DTA^ age-matched mice. B cell development was assayed by plotting absolute counts of total B220^+^, Pre-B, Pro-B, early Pro-B, immature B, and mature B cells in the bone marrow, which is unaltered in *Igg3*^T2ACre:TdTomato^ and *Igg3*^T2A-Cre:DTA^ mice when compared to C57BL/6 mice. We also show Tomato expression is predominantly restricted to mature IgM^+^ and likely re-circulating IgM^–^CD43^+^ B-1 B cells in the bone marrow. The absolute numbers of total CD19^+^, follicular B, marginal zone B, B-1a, and B-1b cells in the spleen and PerC of *Igg3*^T2ACre:TdTomato^ and *Igg3*^T2A-Cre:DTA^ mice is also comparable to C57BL/6 mice. We also provide Tomato expression in these various splenic and PerC B cell subsets.

- A substantial fraction of Tomato+ B cells is negative for IgM and IgG3. Do these cells represent plasma cells, or what else do they represent?

Upon further characterization of these Tomato^+^IgM^–^IgG3^–^ cells, a significant percentage of these cells express CD43, and are likely of B-1 origin. Recent work from Savage et al., 2017, identified a population of CD19^+^CD43^+^IgM^–^ cells, that also spontaneously secrete IgM. We have included a representative flow cytometry plot of CD43 expression on Tomato^+^IgM^–^IgG3^–^ in Figure 1—figure supplement 5D, and we reference the Savage et al. study when discussing these data.

- The subset of Tomato+ B-1a cells should be characterized in more detail (e.g. by analyzing B1-typical markers, responses to stimuli, etc.). Furthermore, given that B-1a cells predominantly reside in the peritoneal cavity, the analysis of Tomato+ cells should not be restricted to the spleen, as it appears to be the case in some figures (such as Figure 2H).

The reviewer importantly points out that the predominant B cell subset in the peritoneal cavity is B-1a cells. However, while only a small percentage of the total B cells in the spleen, B-1a cell numbers are actually comparable in both sites. We, and published work from several other groups, have found that, unlike splenic B-1a cells, peritoneal cavity B-1a cells do not spontaneously secrete IgM. We apologize for not including these data in our original submission, and have added PerC ELISPOT data (comparing Tomato^–^ and Tomato^+^ B-1a cells) to Figure 1G. The rationale behind not including the same time course of Tomato expression in PerC B-1a cells, like we show for splenic B-1a cells in Figure 2H, is that B-1a cells are not detectible in the PerC until about 2 weeks of age. We have revised the main text to better highlight the growing body of evidence that suggests that the spleen is an important site of B-1a activation and spontaneous IgM secretion, which was the rationale behind focusing our analyses on splenic B-1a cells.

- It is an obvious deficit of the paper that Tomato+ cells are not analyzed under GF conditions. This may be technically impossible to achieve, but the authors may comment in the text.

While we agree that this is an exciting future direction, we believe re-deriving the *Igg3*^T2Acre:TdTomato^ reporter mouse in germ-free conditions is currently beyond the scope of this work. We have however added a discussion of this to the main text of the manuscript.

- Some conclusions of the authors are overstatements, such as the title of Figure 4 ("Toll-like receptor signaling is required for B-1a responses to both phosphatidylcholine and the microbiota"). In reality, microbiota reactive IgM and PtC-specific B-1a cells are reduced, but not absent in the TLR knockout animals. Thus, TLRs may be involved but not required. The same applies for the title of Figure 6: Most CDR frequencies in B-1a cells are not affected by the absence of TLRs in the mutant mice.

We apologize for any overstatements, as we agree titles should accurately summarize the findings. The title of Figure 4 has been changed to “Loss of Toll-like receptor signaling results in reduced B-1a responses to both phosphatidylocholine and the microbiota”. The title of Figure 6 has been changed to “B-1a immunoglobulin repertoire analysis reveals unique regulation of a subset of heavy chain genes by distinct subsets of TLRs*”.*

Reviewer #3:

[…] *Some confusion come from the data in which normal amount of IgM was detected in the serum of GF and TLR-/- animals (Figure 3DFigure 4CFigure 5C). Since Tomato+B1a cells are not present in day0 neonatal liver and accumulate upon microbiota colonization (Figure 2F-H), germ free mice are expected to have fewer amount of Tomato+B1a cells. If so, IgM-secreting B1a cells should be fewer in GF and TLR-/- animals. This can be tested by ELISPOT assay for sorted B1a cells in PerC and SPL of GF and TLR-/- mice, like the authors show in Figure 1G. The normal amount of serum IgM gives the impression that there is no defect on Tomato+B1a cells in GF and TLR animals, which gives confusion and disturbance for interpretation of the results related with these animal models.*

Please see our first response to reviewer #1.

*Another confusion comes from the data by showing the* in vitro *down-regulation of CD5 upon TLR stimulations (Figure 7), because the authors suggest that TLR stimulation "induce" the IgM-secreting B1a cells. Especially, Figure 4E directly shows that CD5^+^ B1a cells were reduced in TLR-/- mice. This is confusing, because if TLR stimulation down-regulate CD5 expression, the numbers of CD5^+^ B1a cells should be increased in TLR-/- mice. One possible explanation is that the strength of TLR signal (or amount of the TLR ligands) might be totally different for* in vivo *B1a induction and for* in vitro *CD5 down-regulation. The authors can test this possibility by injecting TLR ligands into WT and TLR-/- mice, and compare the numbers of CD5^+^B1a cells, like in Figure 4E.*

It appears the reviewer may have misinterpreted the data presented in Figure 4E, which actually shows that there is reduction in PtC+ B-1a cells in TLR KO mice. We have spatially reorganized this part of the figure for clarity. We thank the reviewer for this insight, however, since there actually is a trend towards an increased frequency of CD5^+^ B-1a cells in TLR KO mice when compared to WT counterparts (Figure 4B). However, this difference is not significant, which is why we did not feel comfortable originally discussing it in the text. This trend towards an increased frequency of B-1a cells in TLR KO mice indeed fits with our model of TLR stimulation resulting in the down regulation of CD5 on B-1a cells, resulting in “ex-B-1a” B-1b cells. While only speculative, one possible explanation for this difference not being greater is that B-1a cells can regenerate, which may be occurring in WT mice at a more frequent rate to renew the B-1a niche.